# Proteome-scale recombinant standards and a robust high-speed search engine to advance cross-linking MS-based interactomics

Milan Avila Clasen[1,6], Max Ruwolt [2,6], Cong Wang[2], Julia Ruta [2], Boris Bogdanow [2], Louise U. Kurt[1], Zehong Zhang [2], Shuai Wang[3], Fabio C. Gozzo[4], Tao Chen[3], Paulo C. Carvalho [1]✉, Diogo Borges Lima [2]✉ & Fan Liu [2,5]✉

Advancing data analysis tools for proteome-wide cross-linking mass spectrometry (XL-MS) requires ground-truth standards that mimic biological complexity. Here we develop well-controlled XL-MS standards comprising hundreds of recombinant proteins that are systematically mixed for cross-linking. We use one standard dataset to guide the development of Scout, a search engine for XL-MS with MS-cleavable cross-linkers. Using other, independent standard datasets and published datasets, we benchmark the performance of Scout and existing XL-MS software. We find that Scout offers an excellent combination of speed, sensitivity and false discovery rate control. The results illustrate how our large recombinant standard can support the development of XL-MS analysis tools and evaluation of XL-MS results.

Cross-linking mass spectrometry is a powerful technique to analyze protein structures and interactions by providing residue-to-residue connections at low-nanometer resolution[1]. Over the last decade, the scope of XL-MS has expanded from purified proteins/complexes to (sub)proteomes. Two main driving forces of this development were the introduction of MS-cleavable cross-linkers and the development of advanced cross-link search engines[2–5]. Cross-linkers that are cleaved during MS analysis produce linear peptides with signature fragmentation patterns that allow discerning of cross-link mass spectra from others, deriving the mass of the linked peptides and, through further fragmentation, sequencing of the individual peptides. Consequently, the search space increases linearly instead of quadratically with the number of protein sequences in the database, which particularly benefits full proteome analyses and is leveraged by multiple search engines.

Benchmarking XL-MS search output is traditionally achieved by taking three-dimensional protein structures as the ground truth and testing how well they match the identified cross-links[6–8]. However, this approach was shown to substantially underestimate the false discovery rate (FDR) in proteome-wide XL-MS studies[9]. More recently, a synthetic XL-MS peptide standard has been developed to serve as a ground truth for benchmarking XL-MS search engines[10], but this standard only comprises 141 peptides from 38 proteins and thus is too small to provide insights into software performance in proteome-wide XL-MS experiments. Establishing software tools for proteome-wide XL-MS has primarily relied on ad hoc-produced biological samples (for example, fractionated *Escherichia coli* lysate[11,12] and spiked-in [15]N metabolically labeled datasets[13]) or transfer learning of models originally trained on linear peptides[14], but the lack of an analytical XL-MS standard that can mimic complex biological samples has hindered their unbiased validation and benchmarking. An additional challenge for XL-MS software benchmarking arises from the fact that, depending on the specific application of XL-MS, identifications need to be reported on three different levels, referred to as cross-link spectrum matches (CSMs), unique residue pairs (ResPairs) and protein–protein interactions (PPIs).

[1]Carlos Chagas Institute, Fiocruz Paraná, Curitiba, Brazil. [2]Department of Structural Biology, Leibniz-Forschungsinstitut für Molekulare Pharmakologie (FMP), Berlin, Germany. [3]Absea Biotechnology Ltd, ZGC Life Science Park, Beijing, China. [4]Department of Chemistry, Unicamp, São Paulo, Brazil. [5]Charité – Universitätsmedizin Berlin, Berlin, Germany. [6]These authors contributed equally: Milan Avila Clasen, Max Ruwolt. ✉e-mail: paulo@pcarvalho.com; diogobor@gmail.com; fliu@fmp-berlin.de

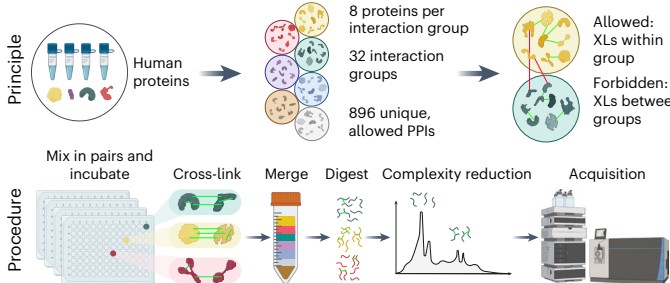

**Fig. 1 | Schematic workflow of the construction of the XL-MS standard.** Proteins were allocated into 32 interaction groups with 8 proteins each. Within the interaction groups, proteins were cross-linked pairwise in all possible combinations, resulting in 28 PPIs per interaction group and 896 PPIs in total. All cross-linked samples were merged before digestion. Created with BioRender.com.

Here we provide an analytical XL-MS standard that is an order of magnitude more complex than existing ones. We produced hundreds of human proteins or protein fragments in *E. coli* and cross-linked them according to a predefined mixing scheme, resulting in fully controlled XL-MS datasets. One dataset guided the development of Scout, an FDR-controlled cross-link search engine that relies on artificial neural networks (ANNs) and is optimized for XL-MS with MS-cleavable cross-linkers. Independent standard datasets were used as a ground truth to benchmark Scout and several state-of-the-art search engines (xiSEARCH/xiFDR[7,15], MaxLynx[16], MSAnnika[4], XlinkX in Proteome Discoverer v.2.5 (ref. 17) (XlinkX PD), MeroX[18]) at CSM, ResPair and PPI levels in differently sized search spaces. The results show that Scout is unique in its ability to unite high sensitivity (high number of true identifications), specificity (accurate FDR) and speed (low processing time). Scout also performed well on previously published datasets for FDR estimation[10,12], yielding more identifications than the best-performing XL-MS search engines in these previous publications. These data demonstrate the power of Scout as well as the potential of our large-scale XL-MS standard to support the development of machine learning-based methods and other algorithmic solutions for analyzing XL-MS data from complex samples.

## Results

### An XL-MS standard mimicking complex biological samples
Our XL-MS standard is based on human proteins or protein fragments produced in *E. coli* (see Supplementary Data 1 for a full list and SDS–PAGE quality control). It is designed to provide a controlled set of allowed and prohibited protein contacts. Since this standard is intended for method development and benchmarking on the liquid chromatography (LC)–MS level, the structural and physicochemical basis of protein contact formation is not relevant. Taking advantage of this fact, we deliberately did not consider biological PPIs. Instead, we divided the proteins randomly into 32 interaction groups with 8 proteins each and cross-linked each group separately with disuccinimidyl sulfoxide (DSSO) under conditions to maximize cross-link formation (Fig. 1). Specifically, proteins from the same interaction group were mixed pairwise in all possible combinations, incubated for 20 min at 50 °C to induce physical contacts, cross-linked and combined into a pooled sample for digestion, strong cation exchange fractionation and LC–MS analysis. Because of the pairwise mixing scheme and the reliability of heat-induced protein contact formation, each protein is interlinked to 7 predefined interactors, allowing up to 896 unique protein pairs (we will refer to them as PPIs for the remainder of this paper because on the LC–MS level they fulfill the same function as biological PPIs). Since we can clearly define which PPIs can and cannot form, this standard serves as a bona fide ground truth to calculate an empirical FDR at the PPI level.

The full analytical standard gives rise to 23,895 tryptic peptides (when considering three missed cleavages, minimum peptide length = 6 amino acids, peptide mass = 500–6,000 Da). Outside of the His-tag, two peptides were shared between three proteins, and 397 peptides (1.66%) were shared between two proteins. On the peptide/ResPair level, it is an additional advantage of heat-induced PPI formation that a wide array of protein conformations and binding interfaces is formed, increasing the probability that all possible lysine–lysine contacts will be stochastically sampled. Therefore, any interlinks between proteins within the same interaction group that a cross-link search engine may identify are highly likely to be true-positive hits, as demonstrated by simulations[19] (Extended Data Fig. 1). Any identified interlinks between proteins from different groups must be false positives. We can also consider any intralinks within proteins as true positives, since, provided that the search space is sufficiently large, they are unlikely to be false positives, as previously shown in experimental datasets[2,12,20] and confirmed in Extended Data Fig. 1. While this degree of certainty does not reach the requirements for a bona fide ground truth for individual cross-links, it is sufficient to make our analytical standard suitable as an experimental benchmark for FDR validation on the CSM and ResPair level.

The analytical standard was split into four batches, each containing eight randomly allocated interaction groups (Supplementary Data 1). Batch 1 was set aside to guide the development of Scout (see 'Developing an ANN-based cross-link search engine'). Batch 2 was used for internal testing to optimize our LC–MS method. Batches 3 and 4 were combined and used for benchmarking Scout and published XL-MS search engines (see 'Benchmarking Scout on independent XL-MS standards').

### Developing an ANN-based cross-link search engine
Our well-controlled XL-MS standard yields datasets that can aid the development of algorithms for XL-MS data analysis. As an example, we introduce Scout, a cross-link search engine for identifying peptides cross-linked with cleavable reagents. Scout is an intuitive, user interface-controlled software that relies on ANNs to generate discriminant functions to score and rank identifications using several quality metrics optimized at the CSM, ResPair and PPI levels. Scout enables multitier FDR filtering at all levels. Scout's workflow is shown in Fig. 2 and described in detail in Supplementary Notes 1 and 2.

We used the batch 1 XL-MS dataset to guide the development of the ANNs (Supplementary Notes 3 and 4). Batch 1 comprises 1,409,900 MS2 spectra, which is similar to proteome-wide XL-MS studies in intact human cells (for example, 1,150,447 MS2 spectra from HEK293T cells, see 'Data Availability' statement). This increases our confidence that the batch 1 dataset can mimic a proteome-wide XL-MS experiment with regard to MS2-level complexity.

### Benchmarking Scout on independent XL-MS standards
In addition to aiding software tool development, the datasets derived from our standard can serve as a ground truth: we know for each detected cross-link whether it is allowed (interlinks and intralinks within one interaction group) or not allowed (between-group interlinks, or intralinks of proteins not present in the respective group) according to our mixing scheme (Fig. 1). This information enables us to calculate an empirical FDR similar to a target-decoy FDR[15], which offers the opportunity for an unbiased benchmarking of XL-MS search engines.

We compared Scout with the following widely adopted XL-MS search engines that are compatible with MS-cleavable cross-linkers: MaxLynx, MSAnnika, XlinkX PD, MeroX and xiSEARCH with the xiFDR module. We benchmarked these tools with two standard datasets (batches 3 and 4) that were used neither during Scout development nor LC–MS method optimization. To avoid that a protein was assigned to the wrong interaction group because of nonunique peptides within

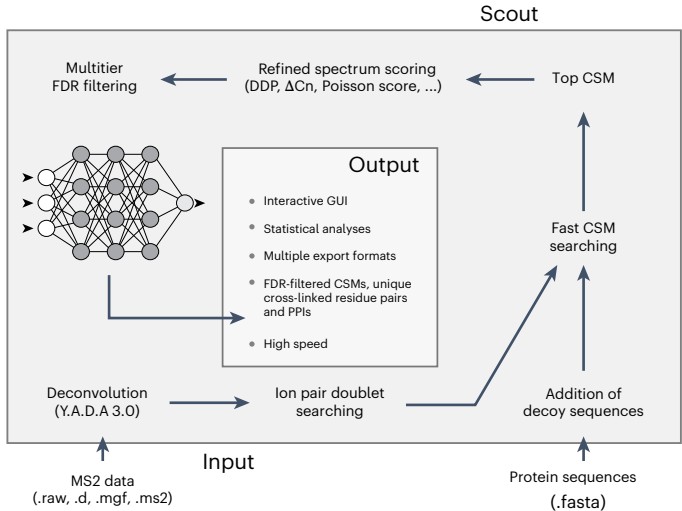

**Fig. 2 | Schematic representation of the cross-link identification workflow employed by Scout.** Scout requires mass spectrometry raw data (MS2 spectra) and a protein sequence database as input. Cross-links are identified in two search steps—ion pair doublet searching and fast CSM searching—which are both described in Supplementary Note 1. The shortlisted peptide pair candidates for each MS2 spectrum are then subjected to refined spectrum scoring based on a set of sensitive quality metrics described in Supplementary Note 2. Finally, the results are filtered according to a user-defined FDR using a machine learning-based discriminant function at each tier of identification: CSMs, ResPairs and PPIs (Supplementary Note 3). The final output is presented through a graphical user interface (GUI), providing a user-friendly display of the identified cross-linked peptides and their associated metrics.

the dataset, we allowed more than one possible group per peptide if the peptide was shared between homologous proteins.

We varied the search space size, using a small 540-protein database and a large 4,000-protein database, both comprising the proteins present in our standard and randomly selected human entrapment proteins from SwissProt with <85% sequence identity to our recombinant proteins (Supplementary Table 1). We selected the same search parameters for each software wherever possible, and set a 1% FDR cutoff within each software (referred to as software-defined FDR) with separate FDR filtering for inter- and intralinks. If a software tool did not provide software-defined FDRs on all identification levels (CSM, ResPair, PPI), we took the software-defined FDR-filtered identifications from the next lowest level and aggregated them to the higher level without any postprocessing filters and score cutoffs (referred to as 'post hoc aggregated results'). A full description of the search settings is provided in Supplementary Table 2. For clearer visualization, results are presented separately for interlinks (Figs. 3 and 4) and intralinks (Extended Data Fig. 2).

Comparing the identification numbers in Fig. 3 and Extended Data Fig. 2, all search engines identified 1–2 orders of magnitude more intralinks than interlinks (on CSM and ResPair level), which is similar to our previously reported numbers for DSSO cross-linking of human cell lysates[2]. For Scout, the fraction of interlinks was ~5% at the CSM level and ~11% at the ResPair level. Interlinks identification by XlinkX PD and xiSEARCH/xiFDR had to be analyzed separately as explained below.

For intralinks, all search engines properly controlled the FDR below 1% on the CSM and ResPair levels, except for MeroX (FDR > 20%). Since intralinks are used for investigating protein structure rather than PPIs, the most relevant information comes from the ResPair level, for which Scout shows the best overall performance (Extended Data Fig. 2).

For interlinks, Scout outperformed MaxLynx, MSAnnika and MeroX on the CSM and ResPair levels, yielding the smallest empirical FDR and highest true-positive identification numbers, irrespective

of the database size (Fig. 3). At the PPI level, MSAnnika reports more true-positive PPIs than Scout, but at a substantially higher empirical FDR (12–15%, Fig. 3, right panels), conceivably because MSAnnika does not include a dedicated FDR control at the PPI level. This emphasizes the importance of controlling FDR at all identification levels[15]. Even at this inflated FDR, MSAnnika only identifies around 300 true PPIs, substantially fewer than the 448 PPIs that are theoretically in the benchmarking dataset (batches 3 and 4) according to our mixing scheme (Fig. 1). Scout, at 1% empirical FDR, identifies 195 PPIs, that is, 43.5% of the theoretical maximum. Such incomplete PPI coverage is also always observed in biological proteome-wide XL-MS datasets, where it results from the relative sparsity and low abundance of cross-linked peptides compared to linear peptides[21].

The interlink comparison of Scout and XlinkX PD was done separately, because the XlinkX PD output is substantially affected by postprocessing settings that are not part of our standard search parameters. In particular, the FDR in XlinkX PD is strongly influenced by heuristic score cutoffs, which were shown to depend on dataset and search parameters[22]. Recent studies made differing recommendations for a static minimum XlinkX score[10,23,24], whereas XlinkX PD developer recommendations indicate that setting a dynamic score cutoff based on the score of the best CSM-level decoy hits in each analysis may be preferable (Methods). We tested both of these options, using either a static score XlinkX cutoff of 60 or a dynamic score cutoff (Fig. 4a,b). We also evaluated how the identification numbers change when increasing the XlinkX score cutoff until 1% empirical FDR is reached (Fig. 4c). On the identification level, Scout outperformed XlinkX PD on CSMs and ResPairs, whereas XlinkX PD identified more true-positive PPIs in most settings. However, on the confidence level, XlinkX PD strongly depends on manually setting suitable score cutoffs during postprocessing, while Scout is able to maintain an empirical FDR < 2% on all identification levels (Fig. 3). Furthermore, XlinkX PD identifications are more susceptible to entrapment and less sensitive in large database searches. At 1% empirical FDR, XlinkX PD reports a higher number of PPIs than Scout when searching against the 540-protein database, but Scout outperforms XlinkX PD in the 4,000-protein database search (Fig. 4c). These results suggest that Scout is more robust and suitable for identifying PPIs against large sequence databases.

The comparisons of Scout to xiSEARCH/xiFDR were also performed separately because searches against the small 540-protein database with xiSEARCH/xiFDR did not run to completion after several weeks when using our standardized settings and server equipment. This is in line with previous reports that proteome-wide applications of xiSEARCH/xiFDR on in-house computers are highly restricted[10]. To still compare Scout to xiSEARCH/xiFDR, a subset of the benchmarking dataset was run on a computer cluster using the 540-protein database and developer-recommended parameters (Methods). Importantly, these parameters included accepting KSTY as cross-linking sites. Therefore, Scout was also run with KSTY specificity in this specific case (other than that the same parameters as in Fig. 3 were used). Scout still reported more correct identifications on all levels when filtering the data with a software-defined 1% or 5% FDR cutoff (Fig. 4d and Extended Data Fig. 3a).

We also compared the data processing times of all tested software by searching our benchmarking dataset (batches 3 and 4) on a computer equipped with 512 GB RAM and powered by dual Intel(R) Xeon(R) Gold 6136 CPUs operating at 3.00 GHz. Scout was substantially faster than the other tools in small and large database searches, and showed the smallest speed decline with increasing database size (Fig. 4e). To be able to compare the processing time of xiSEARCH and Scout using the same computational setup, we limited the searches to four RAW files of the benchmarking dataset and used the default parameters for Scout and xiSEARCH (Extended Data Fig. 3b). Scout processed the data >200 times faster than xiSEARCH. Importantly, while this high-capacity server was needed to meet the RAM demands of some

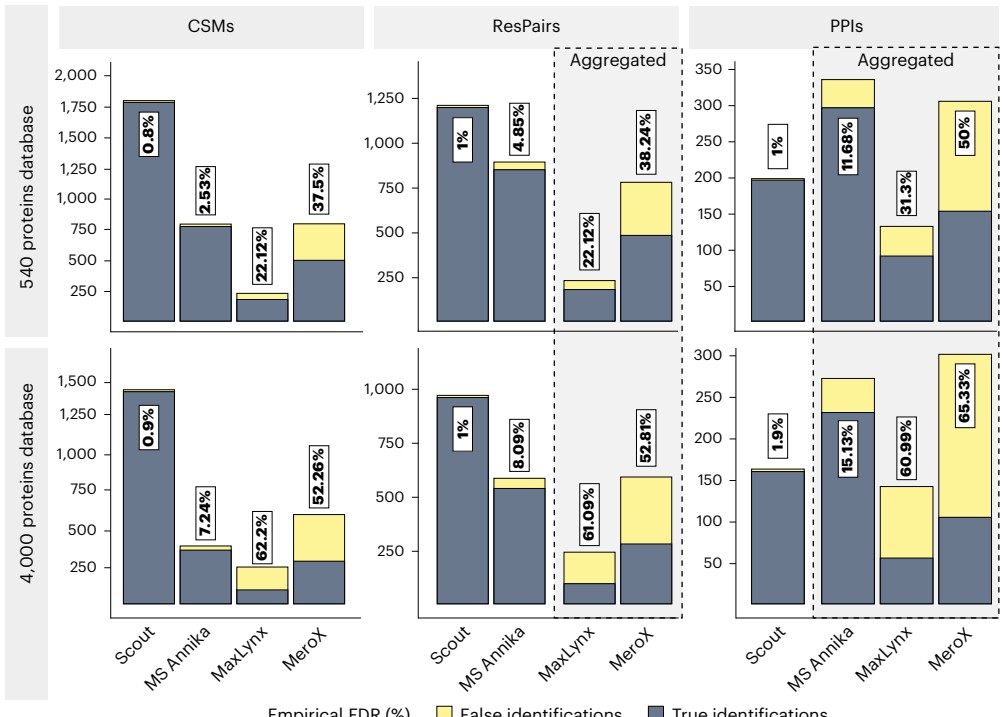

**Fig. 3 | Benchmarking Scout against other XL-MS search engines for interlink identifications.** Number of identified interprotein CSMs, ResPairs and PPIs and the empirically determined FDR at a software-defined 1% FDR cutoff using a 540-protein or 4,000-protein database and identical search parameters, including K as the only cross-linking site. Framed bars mark post hoc aggregated results, that is, cases when CSMs were aggregated to unique ResPairs or unique ResPairs to unique PPIs because search engines do not control FDR at these levels (MeroX and MaxLynx do not report FDR-controlled ResPairs; MeroX, MaxLynx, MSAnnika do not report FDR-controlled PPIs). Blue bars show true-positive identifications, yellow bars show false-positive identifications, violating the mixing scheme of our XL-MS standard.

of the tested tools, Scout operates efficiently with a small memory footprint and is well suited for desktop PCs with as little as 16 GB RAM (see also Extended Data Fig. 4). Thus, Scout provides high sensitivity, specificity and speed on all three levels of XL-MS identifications, irrespective of the search space.

Next, we tested how increasing the search space and adding entrapment sequences impacts the overall performance of Scout when using standard parameters and a 1% software-defined all-level FDR cutoff. Scout maintains a low empirical FDR on all levels at most tested database sizes (Extended Data Fig. 4a). The number of identified interprotein CSMs, ResPairs and PPIs decreases as expected. For example, moving from an entrapment four times higher than the number of experimentally available proteins to an entrapment 160 higher reduces the PPI identifications by 23%. However, our subsequent analyses indicate that this decrease is less drastic than for other search engines (see Supplementary Table 3 and related discussion below). Meanwhile, the search of 55 RAW files with 540 protein entries takes only 2 h on a desktop computer (Intel Core i7 2.90 GHz, 16 GB RAM). When searching the same RAW files against a 35 times larger database (20,622 protein entries), Scout shows an acceptable 4.7-fold processing time increase, to ~9.5 h, demonstrating that it operates efficiently even in large search spaces and is compatible with a standard desktop PC (Extended Data Fig. 4b).

We further evaluated Scout's performance on a published small-scale dataset.[10] Here, a synthetic peptide main library was cross-linked with DSSO according to a mixing scheme. This allows FDR assessments at the CSM and ResPair levels[10]. We compared Scout to the best-performing software reported in the original publication (MSAnnika). Using the same search parameters as above and the database(s) provided in the original publication, we obtained highly similar ResPair-level results for both tools (Fig. 5a). For the

nonoverlapping identifications, however, Scout achieves a lower empirical FDR.

In addition to generating a peptide library, the authors mixed their synthetic peptides with non-cross-linked tryptic HEK cell peptides at a 1:5 ratio to generate a sample with a realistic background of linear peptides, as one would expect from complex interactomics experiments[10]. From this sample, Scout identifies more unique true-positive ResPairs than all search engines reported in the original paper[10], and maintains the lowest FDR and highest identification numbers in searches against databases with 671–20,334 protein entries (Supplementary Table 3). Furthermore, Scout maintains its efficiency and speed in a scenario with higher entrapment: increasing the search space shows that larger database sizes, as expected, reduce the number of ResPair identifications but do not substantially impact the FDR or processing time (Fig. 5b), confirming the results obtained with our own XL-MS standard (Extended Data Fig. 4).

Finally, as the dataset from Matzinger et al.[10] provides only limited PPI-level information, we turned to the dataset from Lenz et al.[12], which aims to provide a PPI-level quasi-ground truth based on the abundance of proteins in cross-linked size exclusion chromatography fractions of *E. coli* lysate. As this dataset was used to advance xiSEARCH/xiFDR, we compared this software's performance to Scout. Scout identifies more PPIs compared to the xiSEARCH result reported in the original publication (Fig. 5c). On all identification levels, both software tools show a low empirical FDR (following the definition from the original publication), but Scout provides more identifications (Fig. 5d).

**Testing Scout on biological XL-MS data**
Finally, we assessed Scout's performance on proteome-wide XL-MS data of biological samples by comparing it to MSAnnika (see also Supplementary Note 5). We first performed an entrapment experiment,

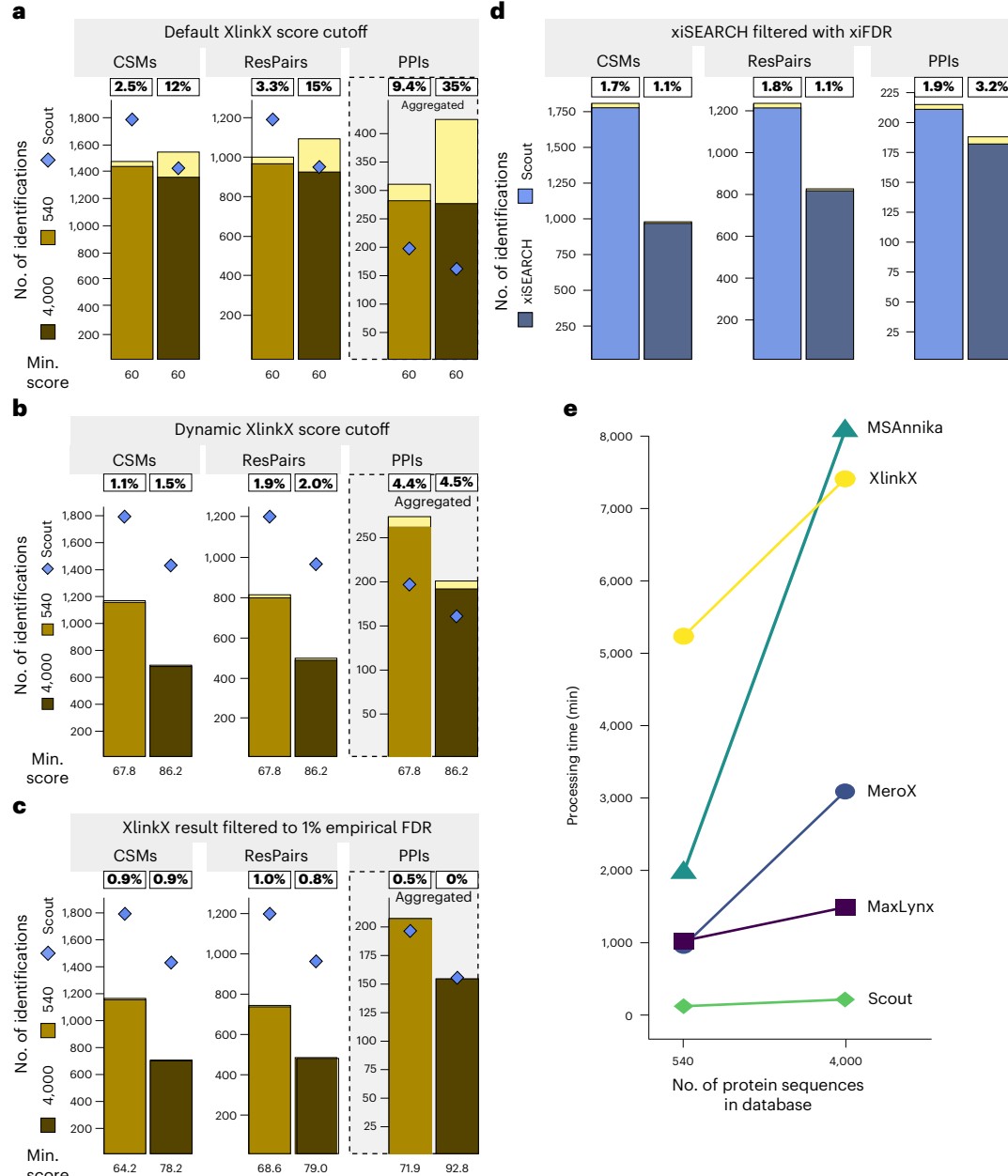

**Fig. 4 | Benchmarking of XlinkX PD, xiSEARCH/xiFDR and software processing times. a–c**, Interprotein CSMs, ResPairs and PPI identifications when comparing Scout and XlinkX PD. True-positive identifications by XlinkX PD are shown in light brown (540-protein database) and dark brown (4,000-protein database). The Scout numbers (blue diamonds) are the same as in Fig. 3. In addition to using our standard search parameters, XlinkX PD identification were postprocessed using a static score cutoff ('default') (**a**), score cutoffs derived from the highest scoring CSM-level decoy in every analysis ('dynamic') (**b**) and score cutoffs set to filter XlinkX PD results to 1% empirical FDR (**c**). The XlinkX score cutoffs are displayed below the bars. Both Scout and XlinkX PD considered K as the only cross-linking site. **d**, Interprotein CSMs, ResPairs and PPI identifications when comparing Scout and xiSEARCH. For Scout, results were filtered at 1% software-defined FDR on all levels. For xiSEARCH/xiFDR, following the developer's recommendation, a 1% software-defined FDR was applied only on

the PPI level using boost between proteins (xiFDR) and reported are the resulting PPIs together with their corresponding CSMs and ResPairs. Scout and xiSEARCH were run using their default parameters, respectively, with KSTY as the possible reaction sites for the cross-linking reagent. In **a–d**, the framed percentage numbers indicate the final empirical FDR and yellow bars show false-positive identifications, violating the mixing scheme of our XL-MS standard. **e**, Processing time in minutes (min) using different search engines on the benchmarking dataset with a 1% software-defined FDR cutoff on a computer with 512 GB RAM and powered by dual Intel(R) Xeon(R) Gold 6136 CPUs operating at 3.00 GHz. xiSEARCH/xiFDR did not run to completion on this hardware setup when using the full benchmarking dataset. Therefore, a separate Scout versus xi speed comparison using only four RAW files was performed and is shown in Extended Data Fig. 3b.

searching a published XL-MS dataset from intact human mitochondria cross-linked with the enrichable, MS-cleavable Azide-A-DSBSO cross-linker[25] against a database containing equal numbers of human mitochondria and *E. coli* proteins (Fig. 6a). Confirming the trends observed in the benchmarking with our standard datasets, Scout

identifies the most CSM and ResPair hits, while providing fewer, but more stringently FDR-controlled, PPI identifications. To illustrate the effect of Scout's PPI-FDR filter, we also report Scout PPI identifications aggregated from the ResPair level (that is, the same approach as used for MSAnnika). The aggregated results of both search

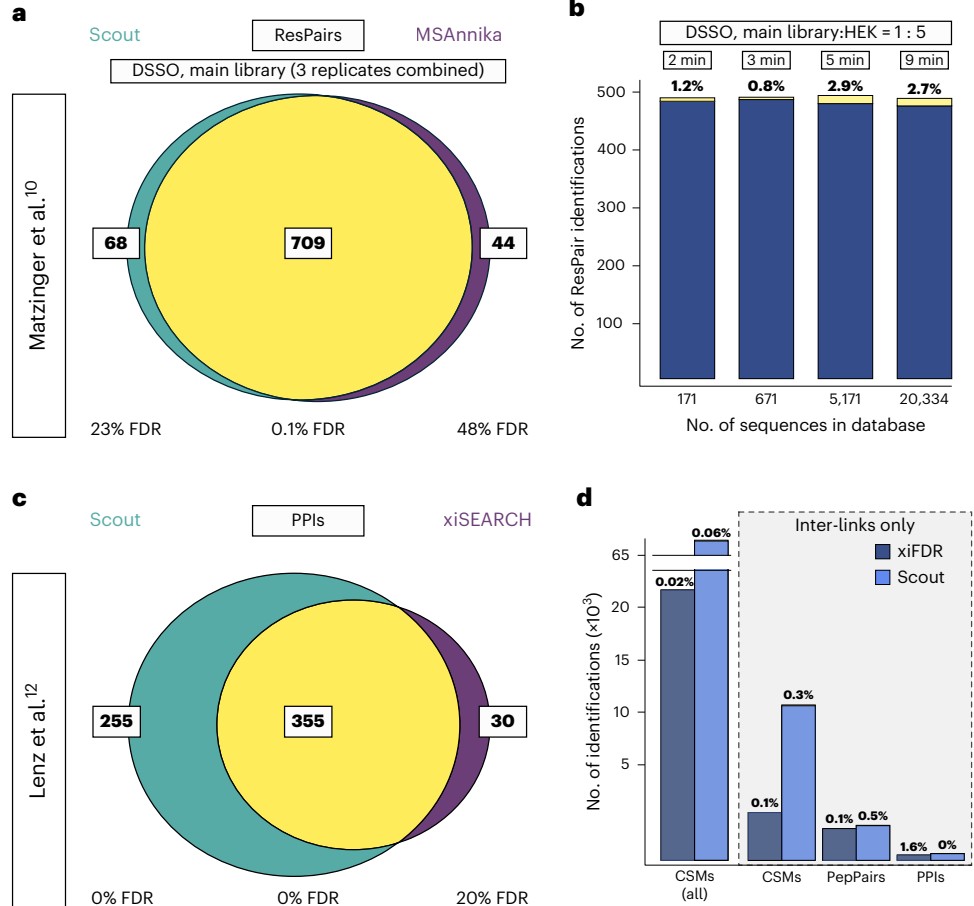

**Fig. 5 | Performance of Scout on published XL-MS benchmarking datasets from Matzinger et al. using synthetic peptides and Lenz et al. using fractionated *E. coli* lysate. a**, Overlap of ResPairs identified by Scout and MSAnnika and the true FDR of Scout-specific (left), shared (middle) and MSAnnika-specific (right) identifications using the DSSO main library from Matzinger et al.[10] and our standard search parameters, which are similar to the ones reported in the original publication. **b**, Scout's true-positive (blue) and false-positive (yellow) ResPair-level identifications from the DSSO main library spiked 1:5 into tryptic HEK peptides when searched on increasingly large databases. The software-defined FDR cutoff was set to 1%; empirical FDR and operating times are indicated above the bars. **c**, Overlap of PPI-level identifications from Scout (left)

and xiSEARCH (right) using the PPI benchmarking dataset by Lenz et al.[12] and a 1% separate software-defined FDR cutoff on the PPI level. Scout was operated with standard parameters and xiSEARCH identifications were retrieved from the original publication. Empirical FDR was determined using the procedure suggested in the original publication.[12] **d**, Performance of Scout and xiSEARCH in identifying intra- and interprotein CSMs, interprotein CSMs only, interprotein PepPairs (peptide pairs) and PPIs when setting an all-level software-defined FDR cutoff of 1%. Scout was operated with standard parameters and xiSEARCH identifications were retrieved from the original publication. The empirical FDR was calculated as described by Lenz et al. and is indicated above the bars.

engines are highly similar in terms of identification number and FDR, showing that stringent PPI-FDR control is a direct consequence of Scout's dedicated filter.

Furthermore, we generated a deep XL-MS dataset from HEK293T cells cross-linked with Azide-A-DSBSO. As there is no available ground truth information, we compared the identified PPIs to STRING[26] and the Negatome database of noninteracting proteins[27] (Fig. 6b). PPI-FDR-controlled Scout identified fewer PPIs than MSAnnika and ResPair-aggregated Scout, but the PPI-FDR filter slightly increased the fraction of medium-to-high confidence PPIs in STRING and reduced the number of Negatome hits. To begin to understand the reasons behind the reduction in PPI identifications when applying the Scout PPI-FDR, we analyzed how many ResPair interlinks support the PPIs found with Scout and MSAnnika. We found that the additional hits in MSAnnika mainly arise from PPIs supported by one ResPair interlink (Fig. 6c), suggesting that Scout's PPI-FDR filter mainly removes PPIs arising from single observations.

To assess the information contained in the ResPair-level identifications, we mapped the Scout and MSAnnika interlinks on

AlphaFold-Multimer models of the PPIs identified by the two search engines (Fig. 6d,e). The structural accuracy of both search engines is highly similar, with MSAnnika performing marginally (2.4%) better on models with >0.5 confidence score (Fig. 6e). However, Scout finds approximately 40% more ResPair interlinks, suggesting that it can provide richer information for structural biology applications.

## Discussion

Advancing methods for proteome-wide investigations critically depends on the availability of thoroughly characterized analytical standards that provide a ground truth to validate new workflows and technologies. This is exemplified by the ProteomeTools project[1], a synthetic peptide library that covers the entire human proteome and important post-translational modifications. ProteomeTools has become an invaluable ground-truth training dataset that has enabled the development of deep learning architectures to predict peptide tandem mass spectra, collision cross-sections and chromatographic retention times[28]. Related efforts for XL-MS have remained at relatively small scale, only covering a few dozen proteins[10], which is too small to

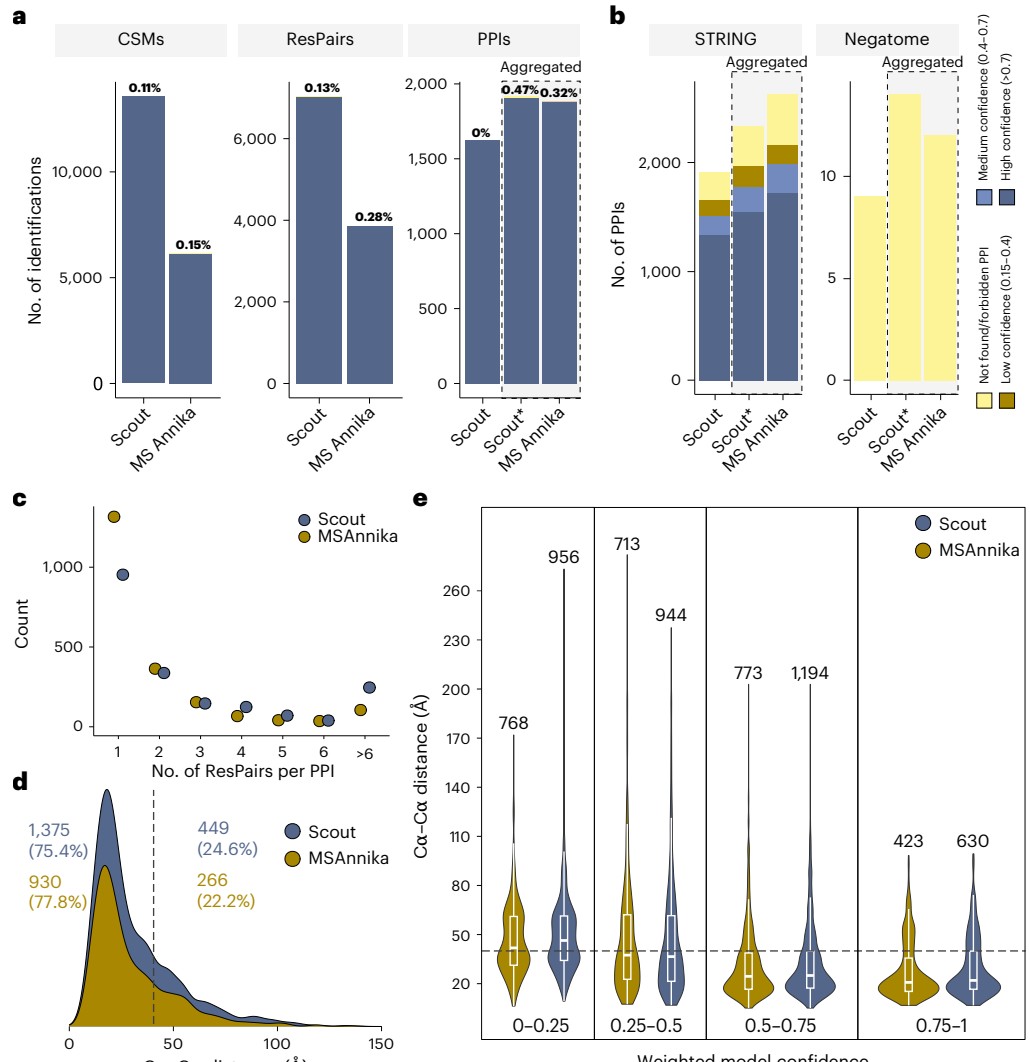

**Fig. 6 | Application of Scout and MSAnnika to biological proteome-wide XL-MS datasets. a**, Entrapment database search on a published dataset of Azide-A-DSBSO cross-linked human mitochondria[25]. The data were searched against 2,000 random human mitochondria proteins sampled from a linear peptide search on the XL-MS data, supplemented with 2,000 random *E. coli* BL21 protein sequences. Interspecies cross-links and *E. coli* cross-links were considered false. Percentages indicate the resulting empirical FDR. **b**, Evaluation of PPIs identified from a HEK cell Azide-A-DSBSO XL-MS dataset. Brown, light blue and dark blue correspond to different STRING confidence score ranges. Yellow represents identifications that could not be found in STRING or that are considered impossible because they match to the Negatome database. In **a** and **b**, PPI-level results for Scout were either determined using the PPI-FDR filter (Scout) or by aggregation of ResPairs to unique protein pairs (Scout*).

The second approach was also used for MSAnnika. **c**, ResPair interlinks per PPIs identified with MSAnnika and PPI-FDR-controlled Scout on the Azide-A-DSBSO HEK dataset. **d**,**e**, Cα–Cα distances of ResPair interlinks identified by Scout (blue) and MSAnnika (brown) when mapped on AlphaFold-Multimer models of their identified PPIs. For each PPI, the model with the highest cross-link satisfaction was used for analysis. Shown are all interlink Cα–Cα distances that can be mapped on AlphaFold-Multimer models with a model confidence of at least 0.5 (**d**), as well as the spread of interlink Cα–Cα distances for different ranges of AlphaFold-Multimer model confidence (**e**). In both cases, only interlinks between residues with a pLDDT score above 50 (indicating an ordered protein region) are considered. Boxes in **e** range from first to third quartile with the median indicated as a horizontal line. Whiskers represent 1.5 times the interquartile range. The violin plot shows that full data distribution, including minima and maxima.

resemble proteomic complexity. Here we mitigate this limitation by formulating a robust XL-MS standard that is an order of magnitude more complex. By systematic mixing and controlled cross-linking of our recombinant proteins, we generated a ground truth of allowed and prohibited cross-link contacts, which results in standard datasets that achieve the MS2-level complexity of published proteome-wide XL-MS studies. The work described in this study took advantage of four standard datasets, but the design of our ground truth makes it readily expandable. While this manuscript was under revision, we created a fifth standard dataset based on eight additional interaction groups. All datasets generated from our ground truth can be seamlessly recombined and reused for future software development and

method benchmarking projects. To support this, they are available as a separate, full PRIDE submission (Methods and 'Data Availability' statement).

To demonstrate the utility of the resulting standard datasets, we used one of them to develop artificial neural network modules for Scout, a new search engine for XL-MS studies with cleavable cross-linkers. Our standard dataset allowed us to optimize Scout's ANNs and avoid overfitting when training them on different identification levels. This also means that the features of Scout are, to some extent, informed by the setup of our XL-MS experiments (Supplementary Note 6). First, Scout is currently limited to cleavable cross-linkers, whereby our analysis of biological data has shown that its applicability

extends beyond the DSSO cross-linker used in our analytical standard. Second, Scout is specifically designed for MS2-based data acquisition strategies (that is, no support for MS3) and has been developed based on raw data from Orbitrap instruments. While Scout can also process data generated with instruments from other vendors (for example, Bruker *.d files and derived MGF files), the absence of a proteome-scale ground-truth dataset measured on other mass spectrometers makes it currently impossible to fairly benchmark Scout's performance on non-Orbitrap data. We envision expanding our analytical standard to other data types in the future to support software testing and development even more broadly.

Our standard datasets offer several distinct advantages over the published peptide-based XL-MS standard[10]. First, cross-linking is performed on proteins in solution and sample preparation for LC–MS is identical to biological samples. Second, composition of the measured sample (linear, intralinked and interlinked peptides) and structure of the resulting data are highly similar to biological XL-MS, which is evident from near-identical MS2-level complexity and intralink/interlink ratios, as well as a comparably incomplete coverage of the theoretically detectable PPIs. Third, and most importantly, the design of our analytical standard allows calculating an empirical FDR at the PPI level, which is not feasible with peptide-based XL-MS standards. Leveraging this advantage, we benchmarked the performance of Scout and other state-of-the-art XL-MS tools on the CSM, ResPair and PPI levels. These results, as well as performance comparisons on published benchmarking datasets[10,12], demonstrate that Scout effectively unites accuracy, sensitivity, specificity and speed. In comparison to other XL-MS search engines, Scout has the shortest runtimes, the highest specificity (low empirical FDR) on all identification levels and the highest sensitivity on the CSM and ResPair levels. On the PPI level, the highest sensitivity is achieved by tools without dedicated PPI-FDR control. In particular, MSAnnika gives high PPI identification numbers for our biological and standard datasets, although the latter show a concomitant FDR increase to 11–15%. This suggests that MSAnnika is well tuned for detection sensitivity, but its PPI identification confidence will be more difficult to control in a real-world scenario without ground truth. XlinkX PD with dynamic score cutoff reaches high PPI sensitivity at a low empirical FDR when searching small-sequence databases, but its sensitivity diminishes with increasing database size. The lower PPI identification numbers of Scout, as well as the even lower numbers of xiSEARCH/xiFDR (the only other available tool with PPI-level FDR control), indicate that losing some PPI detection sensitivity may be a necessary sacrifice to robustly control for false PPIs. Alternatively, Scout's lower PPI sensitivity may be due to the number and mixing scheme of the proteins in our XL-MS standard not fully resembling a natural PPI network, which might not be ideal for the development of ANN-based PPI-FDR control. Hence, tuning the PPI-level performance of Scout and future XL-MS software tools can probably be improved further by designing larger standards with more complex mixing schemes that show greater similarity to PPI networks from 'real-world' biological samples. That said, the currently implemented PPI-FDR filter will still be beneficial when using XL-MS to screen for unknown PPIs with high confidence.

Taken together, the development of Scout and the comprehensive XL-MS search engine benchmarking illustrate how our large-scale XL-MS standard can support the development and testing of next-generation tools and methods to advance proteome-wide XL-MS.

## Online content

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

## Methods

### Protein mixing and cross-linking

Ectopically expressed and purified human proteins were provided by Absea Biotechnology Ltd (Supplementary Data 1) and can be obtained via www.absea.bio. In brief, all proteins are His-tagged at the C termini, packaged in PC3.1 expression plasmid and produced in *E.coli*. Lyophilized proteins were dissolved in 20 mM HEPES pH 7.8 at concentrations varying from 0.1 to 1.5 mg ml$^{-1}$. The dissolved proteins were mixed in pairs of two proteins in all possible combinations within one interaction group. They were incubated for 20 min at 50 °C to induce interactions in vitro. Then, 0.2–1 mM DSSO cross-linking reagent was added to the groups and incubated at room temperature for 30 min. The cross-linking reaction was quenched with 20 mM Tris–HCl pH 8.0 for 30 min at room temperature. Subsequently, all groups were merged, and 8 M urea was added to the solution. The protein mixture was reduced with 5 mM dithiothreitol (DTT) for 1 h at 37 °C, alkylated in the dark with 40 mM chloroacetamide for 30 min at room temperature and digested with 1:200 (wt:wt) Lys-C for 4 h at 37 °C, with shaking. After dilution with three volumes 20 mM HEPES pH 7.8, 1:100 (wt:wt) trypsin was added and incubated overnight at 37 °C. The digestion was stopped by adding 1% formic acid. Peptides were desalted with Sep-Pak C8 cartridges (Waters) and the peptide concentration was determined using the Pierce colorimetric peptide assay (ThermoFisher Scientific). The desalted peptides were dried and stored at −20 °C until further use.

### Strong cation exchange chromatography

Peptides from the benchmarking dataset were loaded in steps of 500 µg onto a PolySULFOETHYL A column (PolyLC) on an Agilent 1260 Infinity II system and separated with a 90-min gradient. All fractions were desalted using C8 stagetips.

### Generation of HEK dataset

We generated a large-scale XL-MS dataset from intact HEK293T cells (CRL-3216, ATCC) using Azide-A-DSBSO (Sigma-Aldrich) as a cross-linker. HEK293T cells were grown to 90% confluence on ten 15-cm cell culture dishes (approximately $8 \times 10^7$ cells) and collected using ice-cold PBS. Then, 20 mg of cells were washed twice with ice-cold PBS before protein concentration estimation using ROTIQuant. Cells were adjusted to a concentration of 10 mg ml$^{-1}$ in PBS and cross-linked in vivo at room temperature using 2 mM Azide-A-DSBSO for 30 min. Quenching was performed with 30 mM Tris–HCl pH 7.4 for 10 min. The cross-linked HEK293T cells were pelleted using a desk centrifuge (room temperature, 10 min, 5,000*g*). The cell pellets were solubilized using 8 M urea in 50 mM triethylammonium bicarbonate (TEAB) pH 8.5 at room temperature and subjected to supersonic treatment using a Bioruptor (5 min, cycle 30/30 s). Then, 1 mM DTT and 0.5 µg benzonase per 10 mg cells were added to the lysate and the sample was incubated for 45 min at 37 °C under shaking at 1,000 r.p.m. Afterwards, the reaction was quenched using 40 mM chloroacetamide and the sample was incubated for 30 min at room temperature in darkness. Endopeptidase Lys-C was added to the sample 150:1 (protein:enzyme, wt:wt) and the sample was incubated for 3 h at 37 °C under shaking at 1,000 r.p.m. The urea concentration was diluted to 2 M using 50 mM TEAB pH 8.5 and tryptic digest was performed overnight (100:1; protein:enzyme, wt:wt).

The digest was desalted using Sep-Pak columns and solubilized in PBS. The isolation of Azide-A-DSBSO cross-linked peptides was performed using DBCO agarose beads (Click-chemistry tools) overnight at room temperature. Afterwards, DBCO agarose beads were successively washed with SDS, 8 M urea in 50 mM TEAB pH 8.5, 10% acetonitrile (ACN) and ultrapure water. The cross-linked peptides were eluted for 2 h using 10% trifluoroacetic acid (TFA) in ultrapure water. Cross-linked peptides were separated from monolinks using size exclusion chromatography on a Superdex 30 Increase column with 3.2 × 300 mm bead dimensions (Cytiva) and the isolated cross-links were subjected to a high-pH HPLC run using a Gemini column (3 µm C19, 110 Å, 100 × 1 mm (Phenomenex) on an Agilent offline fractionation system. The fractions from the high-pH HPLC run were subjected to LC–MS analysis individually.

### LC–MS/MS analysis

Desalted strong cation exchange fractions were resuspended in 1% ACN, 0.05% TFA and 1 µg peptides were injected into a Thermo Scientific Dionex UltiMate 3000 system connected to a PepMap C-18 trap-column (0.075 mm × 50 mm, 3 µm particle size, 100 Å pore size, ThermoFisher Scientific) followed by an in-house packed C18 column for reverse phase separation (Poroshell 120 EC-C18, 2.7 µm, Agilent Technologies) at a flow rate of 300 nl min$^{-1}$. Peptides were separated using a 180-min gradient and analyzed on an Orbitrap Fusion Lumos mass spectrometer with FAIMS Pro device (Thermo Scientific) and Instrument Control Software v.3.4. MS1 and MS2 scans were acquired in the Orbitrap with a mass resolution of 120,000 and 60,000, respectively. The MS1 scan range was set to *m/z* 375–1,600, standard automated gain control (AGC) target, 50 ms maximum injection time and 60 s dynamic exclusion. MS2 scans were set to an AGC target of $1 \times 10^5$, 118 ms injection time, isolation window 1.6 *m/z*. Only cross-linked precursors at charged states +4 to +8 were subjected to MS2. Peptides were fragmented using higher-energy collisional dissociation (HCD) with stepped collision energies (SCEs) of 27 ± 6%. Data were acquired using 2 s per FAIMS compensation voltage (CV) with an internal stepping of CVs from −50 to −60 and −75. The HEK dataset was acquired using the same settings, except for differences in the FAIMS CV combination (−50/−60/−70) and HCD SCEs (19, 25, 30).

### Simulations

Our analytical XL-MS standard is based on the assumptions that: (1) interlinks occurring within one group of eight pipetted proteins are unlikely to arise by chance and are therefore likely to represent genuine true interlinks; and (2) intralinks on proteins physically present in the ground truth are unlikely to occur by chance and therefore also represent genuine true interlinks.

To evaluate both these assumptions, we simulated false residue-to-residue connections in the XL-MS dataset by adopting a recently developed strategy[19]. Briefly, we simulated a distribution of false cross-link matches on the ResPair level, where one site of the linked peptide (α-site) is a true match and the other site (β-site) is a false match. Together, this represents one false connection, that is, a cross-link. First, to simulate matches to the α-site, we calculated the number of cross-links per target protein using our standard datasets (searched with Scout). This was followed by fitting a Zipfian distribution through the ranked target cross-link counts per protein. To simulate false matches to the β-site, we calculated the number of cross-links per decoy protein and fitted a Poisson distribution through the ranked decoy cross-link counts per protein. The fit parameters of both models were utilized to create probability distributions for true and false matches dependent on number of proteins (database size).

We then placed both α- and β-sites onto proteins according to these probabilities in equal number to the decoy cross-link count obtained from Scout output at a given FDR cutoff. The α-sites were placed only onto 256 possible proteins (included in our analytical standard); β-sites were placed onto all entries in the database (540 or 4,000). We then evaluated the count of α- and β-sites co-occurring on the same protein (false intralink) or co-occurring within any of the 32 × 8 randomly assembled groups of proteins (false interlink). This number was then divided by the overall intralink count or the overall within-group cross-link count, as obtained from the Scout output, to give the fraction of false intralinks or false interlinks within the same interaction group that were erroneously accepted as true positives. Of note, we did not consider co-occurrence between two β-sites (which would represent decoy–decoy cross-links) as these events are exceedingly rare compared to α–β co-occurrence.

## Scout

Scout is a user interface-controlled software that should be executed on a computer with a minimum of 16 GB RAM and four computing cores equipped with Windows 10 (64 bits) or later, Python 3.10 or later and the .NET Core 6 or later. Scout has been tested and optimized using Thermo RAW files. The data shown in this manuscript have been generated with Scout v.1.4.14 (https://github.com/diogobor/Scout/releases/tag/1.4.14).

The successor Scout v.1.5 (https://github.com/diogobor/Scout/releases/tag/1.5.0) additionally accepts Bruker timsTOF (.d) files, but has not been comprehensively benchmarked on this type of data. The use of Mascot generic format (MGF) as well as MS2 files is also possible but not recommended, because the content may vary depending on the type of instrument used to record the MS data, as well as the type and settings of the MGF or MS2 converter, which will influence the Scout results.

Scout offers a variety of export formats, report of decoys, graphically annotated spectra, statistical analyses and input tables for xiVIEW[29] and XlinkCyNET[30]. In addition, Scout supports the mzIdentML v.1.2.0 and v.1.3.0 output format, and so offers mzIdentML output for data generated with cleavable cross-linkers. Saving an mzIdentML file, together with a *-specID.ms2 file containing all identified MS2 spectra, enables users to make their data available as a full PRIDE submission. This means that the datasets will be minted a digital object identifier (DOI) in addition to the Proteome Exchange (PXD) accession code, and that data can be directly visualized on PRIDE.

A full description of the Scout workflow, scores and FDR filtering is provided in Supplementary Notes 1–4.

## Data analysis

RAW files were searched for cross-links with multiple search engines: XlinkX node in Proteome Discoverer v.2.5 (Thermo Scientific), MeroX, MaxLynx, MSAnnika, xiSEARCH/xiFDR and Scout v.1.4.14. Search parameters were used as follows: MS1 mass tolerance, 10 ppm; MS2 mass tolerance, 20 ppm; maximum number of missed cleavages, 3; minimum peptide length, 6; peptide mass, 500–6,000 Da. Cross-links were detected by searching for the cross-linker modification on lysines. Carbamidomethylation (+57.021 Da) on cysteines was used as a static modification. Oxidation of methionine (+15.995 Da) was set as a variable modification. Data were searched against databases consisting of the proteins that were used to generate the synthetic dataset and differing numbers of random human entries. Wherever applicable, a software-defined FDR of 1% (on CSM, ResPair and PPI levels) was used to filter data (see Supplementary Table 2 for all search settings). Those search parameters are referred to as 'standard parameters'. For software tools that did not provide software-defined FDRs on all identification levels, software-defined FDR-filtered identifications from the next lowest level were post hoc aggregated to the higher level without any further postprocessing filters and score cutoffs, using the highest reported score of the lower level identifications as the score for the post hoc aggregated higher level identification.

For the benchmarking of XlinkX PD, data were first searched using the software's default settings (minimum score 0, minimum delta 4, software-defined FDR 1% on CSM and cross-link levels). For interlink identification, data were postprocessed by applying either a static score cutoff of 60, (aiming to find a middle ground between previous recommendations for DSSO data[10,23,24]) or a dynamic score cutoff as alluded to by the XlinkX PD developers (https://assets.thermofisher.com/TFS-Assets/CMD/Reference-Materials/pp-001448-ov-proteome-discoverer-presentation-software-application-pp001448-na-en.pdf and ref. 31), where the cutoffs are selected separately for each dataset based on the highest scoring decoy CSMs.

To enable benchmarking of xiSEARCH, a subset of the dataset had to be analyzed on a computer cluster, using the following parameters suggested by the developers: recalibration of the RAW files; MS1 mass tolerance, 3 ppm; MS2 mass tolerance, 6 ppm; maximum number of missed cleavages, 2. Cross-links were detected by searching for the cross-linker modification on lysines. Carbamidomethylation (+57.021 Da) on cysteines was used as a static modification. Oxidation of methionine (+15.995 Da), deamidation of asparagine (+0.983467), methylation of aspartic and glutamic acids (+14.01565) were set as variable modifications. Search results were FDR filtered only on the PPI level using xiFDR v.2.1.5.2. The xiFDR boost function was activated between protein pairs and the remaining settings were left as default.

Standard data were searched against either a 540-protein database or a 4,000-protein database, both comprising the proteins present in our standard and randomly selected human entrapment proteins from SwissProt with <85% sequence identity to our recombinant proteins (FASTA files available through PRIDE, see 'Data Availability' statement). For the proteins in our standard, we included their full-length sequences from SwissProt. As some of our protein constructs are truncated, we considered any identified cross-linked peptides that correspond to protein regions not included in the recombinant protein constructs as false-positive identifications.

HEK cell data were searched against a database generated on the basis of the corresponding proteome determined by bottom-up proteomics, containing 4,860 human protein sequence entries (FASTA file available through PRIDE, see 'Data Availability' statement). Plot generation and the related data processing were done in the R statistical language.

## AlphaFold structure mapping

Unique PPIs of all three datasets were predicted as dimeric protein structures by AlphaFold-Multimer v.2.3 (AF-MM) and the ColabFold infrastructure[32]. We used AF-MM with no templates, a maximum of six recycles, the full database, one as the number of predictions per model and we executed relaxation. Additionally, only protein dimers with cumulatively less than 6,000 residues were modeled. For each PPI (predicted dimeric protein structure), we obtained five ranked predicted structures.

For each interlink in a PPI, the corresponding Cα atom coordinates for both respective residues were extracted utilizing R v.4.4 as well as the bio3d library[33] and the distance was calculated in three-dimensional space over all five predicted ranked models per PPI.

Considering the ranking of predicted structures by model confidence, we extracted and determined the model confidence score as previously defined using the predicted template modeling (pTM) and interface predicted template modeling (ipTM) scores[34]:

$$\text{Model confidence} = 0.8 \times \text{ipTM} + 0.2 \times \text{pTM} \qquad (1)$$

To compare datasets according to their respective identified interlinks, we filtered for ResPair interlinks, in which both linked residues exhibit pLDDT values above 50, to exclude cross-links located in disordered regions.

## Statistics and reproducibility

Recombinant proteins were randomly allocated to interaction groups, which were then randomly split into batches, each containing eight interaction groups.

No statistical method was used to predetermine sample size. No data were excluded from the analyses. The investigators were not blinded to allocation during experiments and outcome assessment.

## Reporting summary

Further information on research design is available in the Nature Portfolio Reporting Summary linked to this article.

## Data availability

The mass spectrometry proteomics data have been deposited to the ProteomeXchange Consortium via the PRIDE partner repository.

The HEK cell XL-MS raw data are available with the dataset identifier PXD043531. The ground-truth raw data used in this manuscript and the corresponding result files underlying the software comparison are available under the dataset identifier PXD052022. The full collection of raw datasets obtained from the recombinant XL-MS standard, the concept of which is reported here, are available as a full submission with the dataset identifier PXD042173 (https://doi.org/10.6019/PXD042173). The published datasets reanalyzed in this study are available through ProteomeXchange with dataset identifiers PXD019120 (fractionated *E. coli* lysate standard dataset from Lenz et al.[12]), PXD029252 (synthetic peptide standard dataset from Matzinger et al.[10]) and PXD046382 (mitochondria XL-MS dataset from Zhu et al.[25]). Source data are provided with this paper.

## Code availability

The Scout software version and source code used for the analyses shown in this study, as well as the latest version of Scout and additional user documentation, are available at https://github.com/theliulab/Scout and https://github.com/diogobor/Scout.

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

## Acknowledgements

We thank L. Mühlberg, Y. Zhu, A. Lewis and T. Bartolec for extensive beta-testing and valuable discussion about features to be added. We are grateful for the advice and support from F. O'Reilly and A. Ciancone on how to operate xiSEARCH/xiFDR. We are indebted to the PRIDE team, specifically Y. Perez-Riverol, who helped us integrate support for the mzIdentML format into Scout. We thank P. Lössl for his help in editing this manuscript. F.L. gratefully acknowledges funding provided by the European Research Council (ERC) Starting Grant (ERC-STG No. 949184) (M.R., C.W. and J.R.), Deutsche Forschungsgemeinschaft (DFG Project LI 3260/6-1) (B.B.), Chinese Government Scholarships (Z.Z.) and Leibniz-Wettbewerb P70/2018 (F.L.). F.C.G. is grateful for the support from Fapesp (2014/17264-3) and National Institute of Science and Technology in Bioanalytics (INCTBio). P.C.C. thanks support grants from Inova Produtos Fiocruz, Fiocruz - PEP, Fundação Araucária - NAPI Proteômica, and CNPq (405934/2002-0, 442655/2023-1 and 310616/2023-9).

## Author contributions

M.A.C., L.U.K., P.C.C. and D.B.L. developed Scout. M.R. designed and prepared the benchmarking dataset. M.R. and D.B.L. analyzed all data. S.W. and T.C. provided recombinant proteins. M.R. and F.C.G. tested the software and assisted selection and optimization of features. C.W. prepared the HEK dataset. J.R. supported the analyses of the biological datasets. B.B. performed the theoretical simulations and provided feedback on the revision. Z.Z. supported the computational analyses and provided feedback on the revision. P.C.C., D.B.L. and F.L. supervised the research. P.C.C. and F.L. provided funding. M.R., F.L., M.A.C., D.B.L. and P.C.C. wrote the manuscript.

## Funding

V. (FMP).

## Competing interests

F.L. is a shareholder and advisory board member of Absea Biotechnology Ltd and VantAI. T.C. is the co-founder of Absea Biotechnology Ltd. S.W. is an employee of Absea Biotechnology Ltd. The remaining authors declare no competing interests.

## Additional information

**Extended data** is available for this paper at https://doi.org/10.1038/s41592-024-02478-1.

**Correspondence and requests for materials** should be addressed to Paulo C. Carvalho, Diogo Borges Lima or Fan Liu.

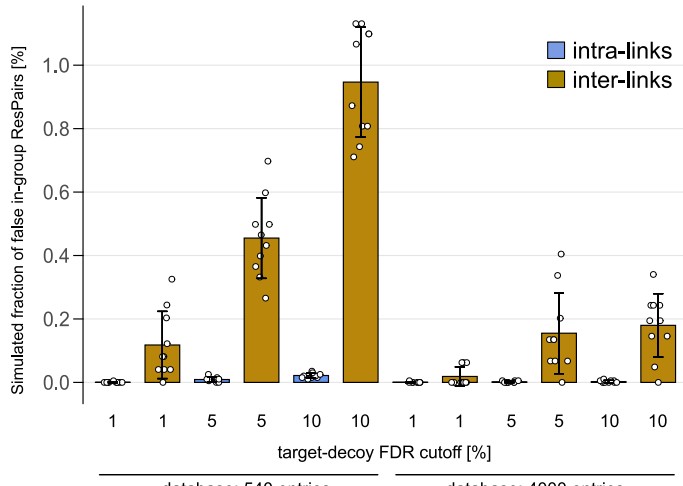

**Extended Data Fig. 1 | Expected fraction of false in-group ResPair cross-links in the XL-MS datasets derived from our recombinant analytical standard.** The results are based on a mathematical simulation approach we recently developed[19]. We considered the 2 database sizes used for our comparison of XL-MS search engines and 3 FDR cut-offs (determined by target-decoy competition in Scout). The bars show the percentage of false ResPair intra-links and inter-links that are expected within the same interaction group, that is cross-links that would be wrongly annotated when defining true-positive hits based on our mixing scheme. Shown are the average +/− SD from 10 simulations with identical input parameters.

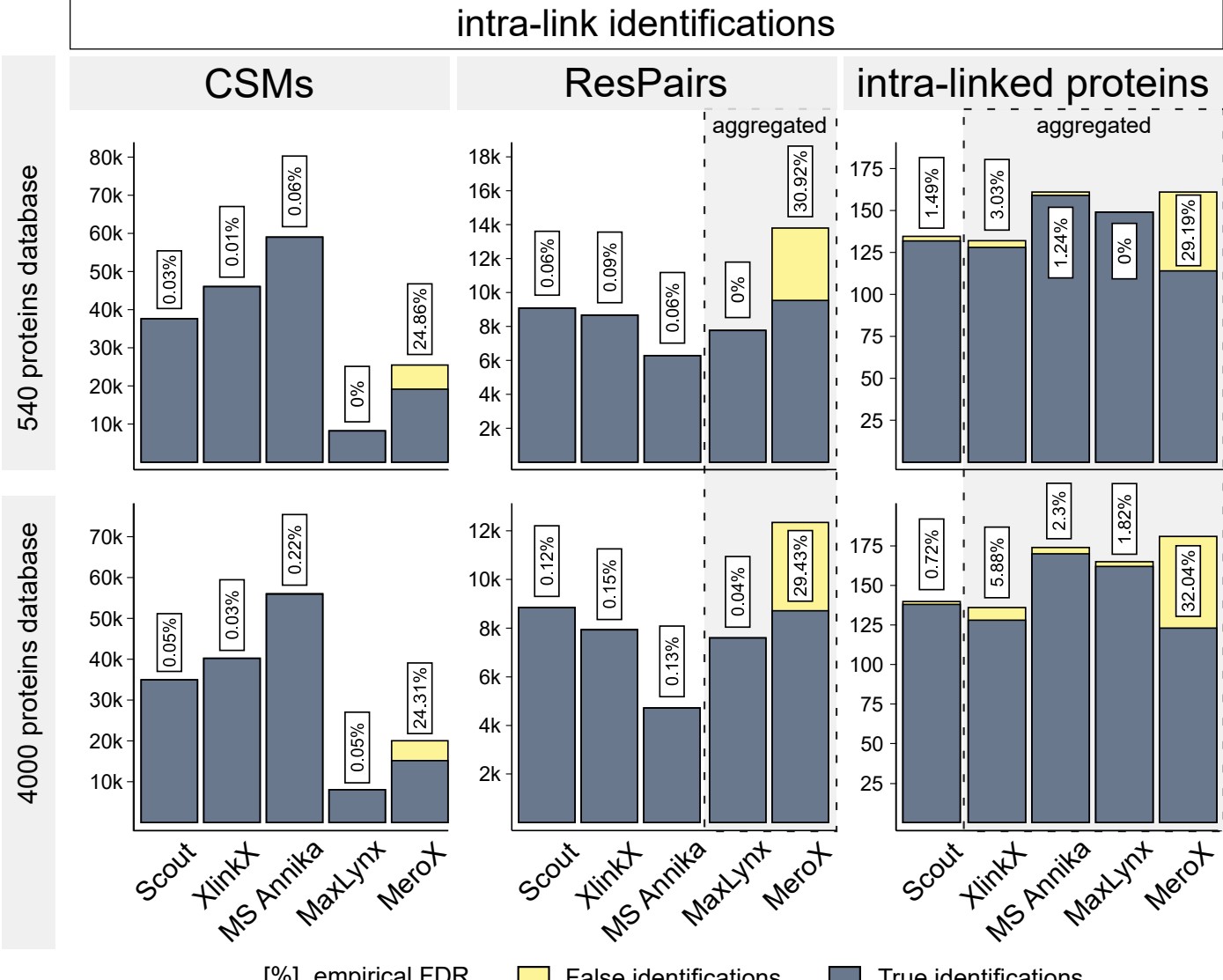

**Extended Data Fig. 2 | Benchmarking Scout against other XL-MS search engines for intra-link identifications.** Number of identified intra-linked CSMs, ResPairs and intra-linked proteins and the empirical FDR using a small (540 proteins, upper panel) or large database (4,000 proteins, lower panel). Framed bars mark post-hoc aggregated results, *that is* when CSMs were aggregated to unique ResPairs or unique ResPairs to unique intra-linked proteins, in case these levels were not directly reported by the software. XlinkX results were obtained with default settings (see Methods) and no further post-processing. The blue bar shows true positive, the yellow bar false positive identifications from which the empirical FDR (shown above the bars) was calculated.

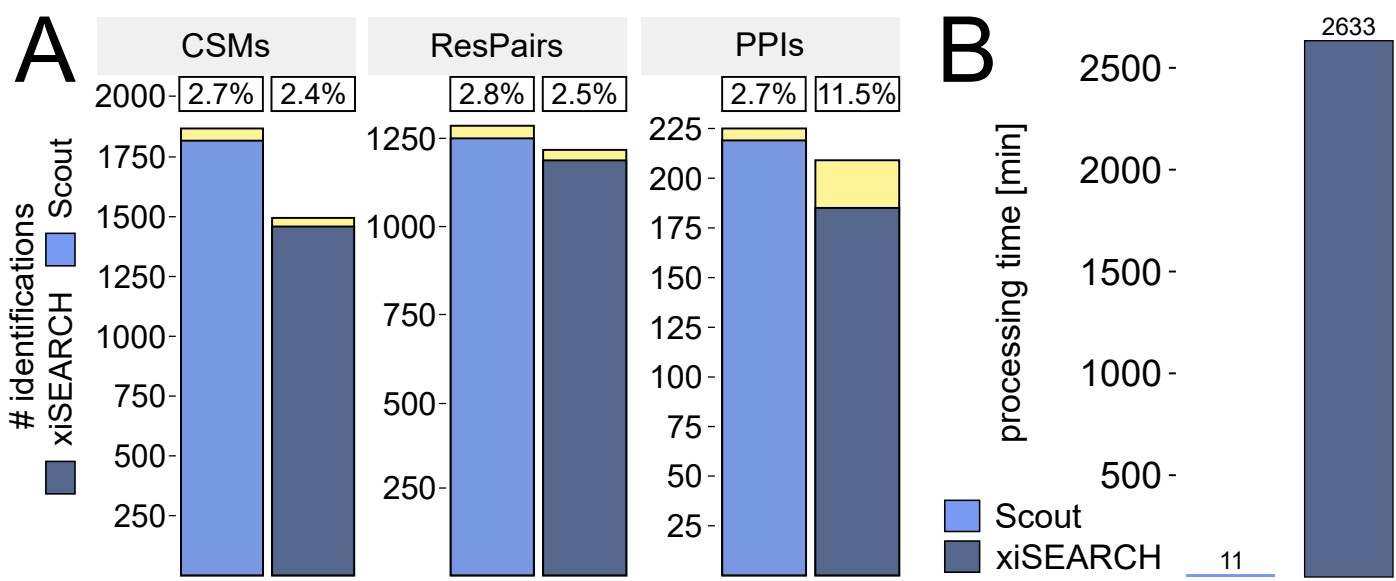

**Extended Data Fig. 3 | Benchmarking of Scout against xiSEARCH/xiFDR.**
(**a**) Inter-protein CSM, ResPairs and PPI identifications at 5% naïve PPI-FDR and empirically determined FDR using Scout and xiSEARCH (both with default parameters, xiFDR with boost between proteins) on a subset of the benchmarking dataset using a 540-protein database and KSTY as possible reaction sites for the cross-linking reagent. (**b**) Processing time of Scout and xiSEARCH on four selected RAW files of the benchmarking dataset on the same computational setup using a 540-protein database.

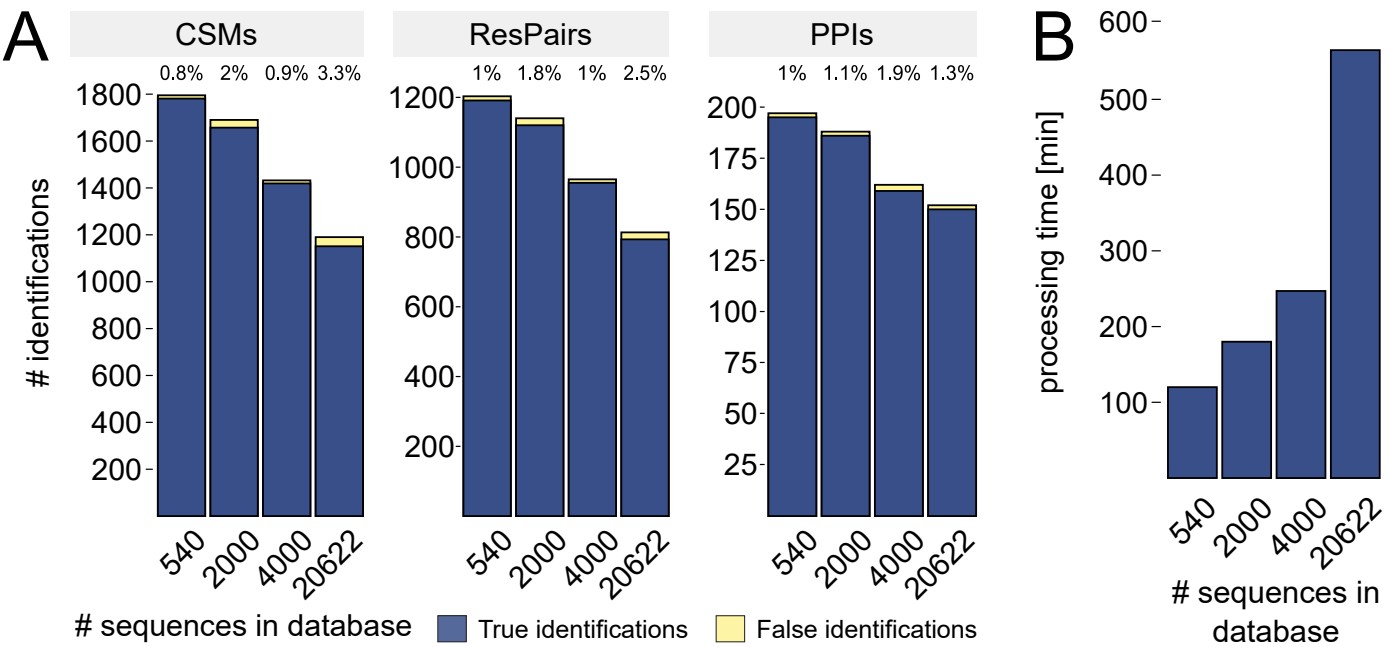

**Extended Data Fig. 4 | Search space-dependent performance of Scout. (a)** Number of inter-protein CSMs, ResPairs and PPIs identified by Scout using standard parameters and all-level 1% naïve FDR cutoff on the benchmarking dataset with increasing database size. True positives shown in blue, false positives in yellow. Empirical FDR is indicated above the bars. **(b)** Processing time of Scout when searching the benchmarking data against increasingly large databases.

# Reporting Summary

## Statistics

For all statistical analyses, confirm that the following items are present in the figure legend, table legend, main text, or Methods section.

| n/a | Confirmed | |
|---|---|---|
| ☐ | ☒ | The exact sample size (*n*) for each experimental group/condition, given as a discrete number and unit of measurement |
| ☐ | ☒ | A statement on whether measurements were taken from distinct samples or whether the same sample was measured repeatedly |
| ☒ | ☐ | The statistical test(s) used AND whether they are one- or two-sided *Only common tests should be described solely by name; describe more complex techniques in the Methods section.* |
| ☒ | ☐ | A description of all covariates tested |
| ☐ | ☒ | A description of any assumptions or corrections, such as tests of normality and adjustment for multiple comparisons |
| ☐ | ☒ | A full description of the statistical parameters including central tendency (e.g. means) or other basic estimates (e.g. regression coefficient) AND variation (e.g. standard deviation) or associated estimates of uncertainty (e.g. confidence intervals) |
| ☒ | ☐ | For null hypothesis testing, the test statistic (e.g. *F*, *t*, *r*) with confidence intervals, effect sizes, degrees of freedom and *P* value noted *Give P values as exact values whenever suitable.* |
| ☒ | ☐ | For Bayesian analysis, information on the choice of priors and Markov chain Monte Carlo settings |
| ☒ | ☐ | For hierarchical and complex designs, identification of the appropriate level for tests and full reporting of outcomes |
| ☒ | ☐ | Estimates of effect sizes (e.g. Cohen's *d*, Pearson's *r*), indicating how they were calculated |

*Our web collection on statistics for biologists contains articles on many of the points above.*

## Software and code

Policy information about availability of computer code

| Data collection | Orbitrap Fusion Lumos Instrument Control Software version 3.4 (Thermo Scientific) |
|---|---|
| Data analysis | Scout v1.4.14 (source code, software and user documentation available at https://github.com/theliulab/Scout). Data were also searched with xiSEARCH/xiFDR v2.1.5.2, MaxLynx, MSAnnika, XlinkX in Proteome Discoverer v2.5 (Thermo Scientific), and MeroX - references to the original publications are provided in the main manuscript. |
| | R v.4.4, AlphaFold-Multimer 2.3, ColabFold (Mirdita, M. et al. ColabFold: making protein folding accessible to all. Nat Methods 19, 679-682, 2022) |

For manuscripts utilizing custom algorithms or software that are central to the research but not yet described in published literature, software must be made available to editors and reviewers. We strongly encourage code deposition in a community repository (e.g. GitHub). See the Nature Portfolio guidelines for submitting code & software for further information.

# Data

Policy information about availability of data

All manuscripts must include a data availability statement. This statement should provide the following information, where applicable:
- Accession codes, unique identifiers, or web links for publicly available datasets
- A description of any restrictions on data availability
- For clinical datasets or third party data, please ensure that the statement adheres to our policy

TThe mass spectrometry proteomics data have been deposited to the ProteomeXchange Consortium via the PRIDE partner repository:
- The HEK cell XL-MS raw data are available with the dataset identifier PXD043531
- The ground-truth raw data used in this manuscript and the corresponding result files underlying the software comparison are available under the dataset identifier PXD052022
- The full collection of raw datasets obtained from the recombinant XL-MS standard, the concept of which is reported here, are available as a full submission with the dataset identifier PXD042173
The published datasets re-analyzed in this study are available through ProteomeXchange with dataset identifiers PXD019120 (fractionated E.coli lysate standard dataset from Lenz et al.12), PXD029252 (synthetic peptide standard dataset from Matzinger et al.10), and PXD046382 (mitochondria XL-MS dataset from Zhu et al.25).

# Research involving human participants, their data, or biological material

Policy information about studies with human participants or human data. See also policy information about sex, gender (identity/presentation), and sexual orientation and race, ethnicity and racism.

| | |
|---|---|
| Reporting on sex and gender | n/a |
| Reporting on race, ethnicity, or other socially relevant groupings | n/a |
| Population characteristics | n/a |
| Recruitment | n/a |
| Ethics oversight | n/a |

Note that full information on the approval of the study protocol must also be provided in the manuscript.

# Field-specific reporting

Please select the one below that is the best fit for your research. If you are not sure, read the appropriate sections before making your selection.

☒ Life sciences  ☐ Behavioural & social sciences  ☐ Ecological, evolutionary & environmental sciences

For a reference copy of the document with all sections, see nature.com/documents/nr-reporting-summary-flat.pdf

# Life sciences study design

All studies must disclose on these points even when the disclosure is negative.

| | |
|---|---|
| Sample size | The size of the recombinant protein standard was chosen such that a MS2-level complexity comparable to published proteome-wide XL-MS studies could be achieved. Samples size for the HEK cell XL-MS experiment was chosen based on preliminary experiments and common practice in the field; without any statistical sample size calculation. |
| Data exclusions | No data were excluded from our analyses. |
| Replication | Replication is not directly relevant because the focus of this work was to obtain a large-scale ground truth dataset based on a newly developed recombinant XL-MS standard. We used this ground-truth data to assess the reproducibility and robustness of existing XL-MS search engines and develop the Scout search engine. The reliability of Scout was independently validated using a previously published small-scale ground-truth dataset, showing that the vast majority of Scout identifications agree with the identifications of the best-performing software in the previous publication (Matzinger et al., Nat Commun, 2022). Replication is also not directly relevant to the biological XL-MS data (re-)analyzed in this study because they were only used to confirm the functionality and performance of published XL-S search engines and Scout. |
| Randomization | The proteins in our recombinant XL-MS standard were randomly allocated to interaction groups, each containing 8 proteins. The interaction groups were randomly split into batches, each consisting of 8 interaction groups. Randomization was not relevant to the other experiments because they did not involve any group allocation. |

| Blinding | Not relevant to this study. In order to generate a ground-truth dataset, investigators needed to know which proteins are in which interaction group. |
|---|---|

# Reporting for specific materials, systems and methods

We require information from authors about some types of materials, experimental systems and methods used in many studies. Here, indicate whether each material, system or method listed is relevant to your study. If you are not sure if a list item applies to your research, read the appropriate section before selecting a response.

## Materials & experimental systems

| n/a | Involved in the study |
|---|---|
| ☒ | ☐ Antibodies |
| ☐ | ☒ Eukaryotic cell lines |
| ☒ | ☐ Palaeontology and archaeology |
| ☒ | ☐ Animals and other organisms |
| ☒ | ☐ Clinical data |
| ☒ | ☐ Dual use research of concern |
| ☒ | ☐ Plants |

## Methods

| n/a | Involved in the study |
|---|---|
| ☒ | ☐ ChIP-seq |
| ☒ | ☐ Flow cytometry |
| ☒ | ☐ MRI-based neuroimaging |

## Eukaryotic cell lines

Policy information about cell lines and Sex and Gender in Research

| Cell line source(s) | HEK293T (CRL-3216, ATCC) |
|---|---|
| Authentication | Cell line was not further authenticated. |
| Mycoplasma contamination | Cell lines was tested negative for mycoplasma contamination. |
| Commonly misidentified lines (See ICLAC register) | No commonly misidentified lines were used in this study. |

