## [Peer Review File · Nature Methods]

Proteome-Scale Recombinant Standards and a Robust High-Speed Search Engine to Advance Cross-Linking MS-Based Interactomics

Corresponding Author: Professor Fan Liu

A version of this paper was originally rejected for publication by Nature Methods, however that decision was reconsidered after appeal by the authors.

Version 0:

Decision Letter:

7th Dec 2023

Dear Professor Liu,

Your Article entitled "Proteome-Scale Recombinant Standards and a Robust High-Speed Search Engine to Advance Cross-Linking MS-Based Interactomics" has now been seen by 3 reviewers, whose comments are attached. While they find your work of potential interest, they have raised several concerns which in our view are sufficiently important that they preclude publication of the work in Nature Methods, at least in its present form.

As you will see, the reviewers raise concerns about differences in how the various approaches were compared to Scout, the lack of sufficient discussion about your performance evaluations as well as missing details regarding the method.

Should further experimental data allow you to fully address these criticisms we would be willing to look at a revised manuscript (unless, of course, something similar has by then been accepted at Nature Methods or appeared elsewhere). This includes submission or publication of a portion of this work somewhere else. We hope you understand that until we have read the revised paper in its entirety we cannot promise that it will be sent back for peer-review.

Although we cannot publish your paper, it may be appropriate for another journal in the Nature Portfolio. If you wish to explore the journals and transfer your manuscript please use our manuscript transfer portal. You will not have to re-supply manuscript metadata and files, unless you wish to make modifications. For more information, please see our [manuscript transfer FAQ](http://www.nature.com/authors/author_resources/transfer_manuscripts.html?WT.mc_id=EMI_NPG_1511_AUTHORTRANSF&WT.ec_id=AUTHOR) page.

If you are interested in revising this manuscript for submission to Nature Methods in the future, please contact me to discuss your appeal before making any revisions. Otherwise, we hope that you find the reviewers' comments helpful when preparing your paper for submission elsewhere.

Sincerely,
Arunima

Arunima Singh, Ph.D.
Senior Editor
Nature Methods

Reviewers' Comments:

Reviewer #1:

Remarks to the Author:

In their manuscript, the authors describe a mix of synthetic proteins as new benchmarking tool to validate FDR of cross-linking MS software at PPI level that is designed to mimic complex samples as whole cell cross-link experiments. It's therefore a nice complementary tool for benchmarking on PPI level compared to existing peptide-libraries that allow to validate FDR on residue pair level. The authors use the synthetic protein mix to develop their own novel XL-MS search engine, Scout. Scout delivers excellent results by means of identified XLs and PPIs while maintaining a low FDR as shown by usage of a published peptide library and their own protein mix. Full transparency is guaranteed as all raw files are made available via PRIDE and the code is made available on github.

With no doubt the new software, as well as the datasets created on the synthetic peptide mix will be of value for the community, especially as the software is free of charge, easy to use and seems to produce reliable results. This reviewer is concerned though, if the current work is suitable for Nature Methods due to the lack of an application of the new method to a biological question demonstrating its advantage over existing approaches. Such an application could however be added to a revised version of this manuscript, even the reanalysis of existing XL studies finding new insights with Scout would demonstrate its applicability to a biological question.

Please find below my comments and questions to the current manuscript:

Scout: In the hands of this reviewer Scout worked out of the box. It comes with an intuitive to use GUI that allows to adopt search parameters, define custom linker reagents, enzymes, or modifications. It works only with MS cleavable linkers, but for the aimed usage of proteome wide searches it makes sense to focus on such linker types. Furthermore, Scout offers export functionalities for post processing. Most importantly, Scout is indeed very fast and needs less resources compared to other commonly used software and gives a reliable FDR which makes it an excellent choice for proteome wide analyses using MS cleavable linkers! For direkt software-internal visualization, Scout comes with a very nice spectrum viewer and outputs some helpful plots for statistics and QC.

Here some suggestions for improvement:

- A direct export for usage of XlinkCyNet, a data visualization tool from the author's lab, is possible. Direct support for more visualization tools such as xiView would even improve the usability for a broad range of users here.
- While all works fine with Thermo raw-files and this reviewer obtains similar numbers as shown in the manuscript with some sample files, the analysis of Bruker (.d) files seems problematic. When loading a ".d" file, Scout crashes in my hands and when loading the converted mgf instead, it gives only ~1/5 of the crosslink IDs compared to MaxLynx. Did the authors test to analyze Bruker files themselves and can report their results in an updated manuscript? This would certainly further improve its quality and would broaden the potential user-group.

Synthetic protein mix:

The authors created a mix of 32 groups of 8 proteins each resulting in close to 900 true positive PPI hits. This allows to experimentally validate FDR on PPI level, while FDR on residue pair or spectrum match level can only be estimated. In this study all intra-protein crosslinks are considered as true positive, which however is only an assumption and not a real ground truth. This is important to mention also in the discussion of this study.

- To build the library both proteins and protein fragments were used. I was wondering if there are cases of crosslinked residue pairs to sequence parts of proteins not existing (meaning that they might be counted correct but since only a protein fragment is present in the sample the software might output a crosslink to a part of the protein that is not part of the synthesized fragment). Have you checked on that and have you included full protein sequences into the FASTA files of your searches or only the existing fragments?
- What's the share of homologous peptides across the protein groups used for crosslinking?

Further comments:

- Supplementary Table 2: The search parameters of the benchmarked software tools are not equal, most importantly most tools were set to search links from K to K but Scout is allowed to search for links to K,S,T and Y. This seems not fair for comparison as Scout is "allowed" to find way more combinations for unique residue pairs. Please either explain why parameters were chosen like that or correct for harmonized parameters across all tested search engines. In addition, the number of allowed miscleavages is lowered to 2 for Xi while its at 3 for all others, why? Was this done to reduce needed search space (as the authors had to deal with very long runtimes for Xi)?
- Figure 3/4: On PPI level Scout delivers excellent FDR control, however as already mentioned by the authors themselves, a direct comparison is not possible as Annika and XlinkX do not report PPIs, hence do not have FDR control on that level.
- The comparison to Xi makes sense and Scout reports more PPIs at lowered FDR (Figure 4 panel D). However, I was wondering about the following:
 - Why the processing time of Xi is not shown in Figure 4 panel E (but only in a supplementary figure). This would highlight the main advantage of Scout over Xi?
 - Furthermore, I didn't fully understand why the ID numbers differ here for Scout:
 - o Figure 3/540 protein FASTA: ~200 PPI at 1% empirical FDR
 - o Figure 4D/540 protein FASTA: ~210 PPI at 1.9% empirical FDR
 - o Supplemental Figure 2A/540 protein FASTA: ~ 225 PPI at 2.7% empirical FDR
 - And for Xi:
 - o Supplemental Figure 2A/540 protein FASTA: ~200 PPI at 11.5% empirical FDR
 - o Figure 4D/540 protein FASTA: ~180 PPI at 3.2% empirical FDR
- This reviewer suggests to re-design Figure 4 to enhance clarity for the reader.
- The addition of a list that defines which protein in which group in order for others to validate/use the dataset would improve broad application of this "protein library" as a tool for the community.

Reviewer #2:

Remarks to the Author:

The present work provides a very important and highly useful dataset + software for the community. The ability to estimate PPI error is unique and the ability to control it explicitly is also helpful (and not standard across different softwares). Software is impressively fast, and it is clear that care has been taken to ensure accuracy of the identifications at all levels of cross-link identification. I am overall very enthusiastic about the paper which will have a substantial impact on the crosslinking field and would suggest publication after the comments below are addressed.

Line 43 - Reference 6 is probably not the most appropriate citation for this statement, as it is only mentioned briefly and does not contribute substantially to their quality control filtering of the dataset. Reference PMID:30649854 explicitly (and successfully) applied structural mapping to segregate correct and incorrect crosslink spectral matches. Other papers also do this as it is the convention in the field. I suggest to either (1) cite more large-scale XL-MS papers that perform some structural mapping as benchmarking, or (2) reword the statement to say something to the effect of: One way that XL-MS search outputs can be assessed is by....

Line 81 - suggest explicitly mentioning how cross-link production was maximised in the main text here (20 min incubation at 50C to cause aggregation before crosslinking). On a related note, it would be useful to mention how many of the potential 896 PPIs you were able to detect crosslinks for, using your aggregation approach. Were there any known bona fide PPIs present in the mix?

Line 89 - ref 31267744 should also be cited. Also, the text should be tweaked - the fact that intra-protein XLs are mostly always true-positives is only really true for large scale XL-MS studies. Smaller databases (such as those used in small scale studies) matched to many cross-linked spectra may lead to a higher proportion of false-positive intralinks, as identifications are forced to match those proteins.

Line 107 - How were these CSMs identified? Which program/s? It is not clear after reading both the main text and the methods section, and I was not able to identify which dataset this was in the PRIDE submission. Overall, I think there needs to be more statistics describing the results and composition of this synthetic training/evaluation dataset better.

Line 107 - In line 89, you state that most erroneous IDs in XL-MS searches stem from interlink cross-links (as bad peptide matches are unlikely to both map to the same protein by chance in a large database). Given you want to model the error contributing to this high rate of inter-link false-positives by generating a synthetic dataset with PPIs, it would be important to explicitly report here how many CSMs you had for inter vs intra. I would guess that even with the 50C aggregation step, a large proportion of CSMs from your synthetic dataset would be intra-links. If the aggregation step was able to substantially increase the proportion on inter-link peptides and hence enrich your synthetic dataset with these species, this is worth emphasizing in the main text as a significant advantage of your dataset over others already published.

Line 131 - it would be clearer to mention the number of "standard" datasets used, which is, if I understood correctly, "... benchmarked with the remaining 24 protein standard groups."

In Figure 3, and related text, the terminology aggregated is used to describe the collapsing of error across redundant levels of XL-MS error (CSM to ResPairs to PPIs). Not necessary but, for clarity's sake and to help the reader, I would suggest a different terminology to clarify that these programs do not intrinsically control the FDR to these levels (hence the problem, and on the other hand, a key benefit of your program). For example, "manually collapsed", or "post-hoc aggregated", etc.

Line 171 - The yeast cross-linking study used to define "default" XlinkX settings used in Figure 4A suggested a two-score insearch cutoff to control PPI FDR in XlinkX - not just a minimum of XlinkX score 60 as used in this study, but also a minimum delta XlinkX score of 10 (Figure 4H of PMID:31851481). I would suggest to report either the results (1) from settings as optimised and reported in that study (minimum XlinkX score of 60, and delta score of 10), or (2) as from the default Proteome Discoverer 2.5 XlinkX settings with just 1% FDR settings in search, or (3) consider omitting and use just "dynamic" and empirically filtered XlinkX results (Figures 4B and C). The yeast study used an earlier version of Proteome Discoverer and hence version of XlinkX (which updates alongside with each Proteome Discoverer update and where each update performs different FDR control strategies). Therefore, I would suggest reporting either of the last two options would be most appropriate.

Line 266 - It is interesting to see that, although the FDR is impressively low in both Scout and xiSEARCH PPI identifications, there is a relatively small amount of overlap. It would be useful to have more discussion on why this may be and pointing out that the softwares can be very complementary as a discussion point - what factors about the spectra, peptides or proteins (for example, abundance or more complex modified peptides) are each of these programs missing? Another reason could be that they are they missing well-formed doublets required for Scout identification, but not for xiSEARCH identification. Some analyses of these Scout-unmatched spectra would be very useful and informative.

More general comments:

Is there compliance with export of data in mzIdentML 1.3.0, for PRIDE submissions?

The authors need to more explicitly describe which type of MS analyses are compatible, and also performed at different steps. It seems here that software was trained on, and analyses stepped-HCD MS/MS of DSSO cross-links. Does it work for other MS-cleavable cross-linkers? Does it work for MS3 analysis? How well does it generalise - e.g. the efficiency of doublet

generation is different across different crosslink molecules.

Also would be good to have a forward looking discussion of possibility of expanding Scout to other types of crosslinkers.

As stated above, this is a very impressive and important study, yet it would be very nice if a completely new “real life” Xlinking dataset would be included in the study in addition to the analysis previously published datasets.

Regarding NN architecture. From what is written in Supp Note 4, the neural network is a canonical Multi-Layer Perceptron. Its main purpose is to capture non-linearities between inputs that are relevant to predict the output. Here, the output is a binary state indicating whether two proteins really interact or not. Overfitting to the training data is the most important aspect to consider in this type of densely connected neural nets. To prevent it, authors use two approaches: 1) Fine tune the model to a subset of 64 proteins in their standard and 2) L2 regularization on neurons weights. I am curious about how these 64 proteins were selected, as this is not indicated in any part of the manuscript. The second point that raises my curiosity is why they do not provide any standard metric about the binary classification performance during training following a cross-validation scheme (e.g. AUROC, or in this case AUPRC if TP/TN ratio is really small).

Reviewer #3:

Remarks to the Author:

This manuscript presents both a new approach to generate a large-scale XL-MS benchmark dataset and a new XL-MS search engine. The scale of the benchmark dataset and performance of Scout even on already existing benchmarks looks good at first glance. The manuscript is well-written but a number of important details are missing that would help properly evaluating it. Scout appears better than a number of search engines used in the field although XlinkX and xiSEARCH come close on the PPI level, once FDR is properly controlled. Scout is also much faster, however a number of questions and concerns are raised below about the details of performance comparisons.

1) The authors propose that instead of peptide synthesis as described in Matzinger et al, to express whole proteins or fragments in E.coli. It's important to note that both approaches seem to produce similarly useful but unrealistic datasets. Rather than cross-linking purified peptides, here the authors cross-link purified proteins after heating them to induce denaturation/aggregation. Since this has no biological relevance, the main advantage seems to be efficiency/scale of peptide production? It might help the reader to explain the general thinking behind the approach compared to just synthesizing more peptides, considering the more predictable results of peptide synthesis. The nature of “PPIs” produced this way seems about as artificial as mixing subsets of purified peptides. In fact, I would be much more comfortable labeling the data resulting from the benchmark experiment something like “Protein Pairs” throughout the manuscript to avoid confusion with actual interactors in other datasets.

2) While the scope of the experiment described in Figure 1 appears impressive, digging into the details, it seems only a fraction of the inter-protein cross-links can be detected. Of the 896 possible protein pairs, 193 are detected in searches of batches 3 and 4, using the 540 fasta, which is presumably the most sensitive. I don't see any information to help figure out what fraction of the overall dataset this represents, but since ¼ of the dataset was used for training, and there are 4 batches mentioned, I would guess this represents ½ the dataset? If that's the case, the authors are detecting ~40% of possible cross-linked protein pairs? This would be an important metric to report.

One possible explanation for such relatively low rate could be that proteins are purified “at concentrations varying from 0.1–1.5 mg/mL” That's a pretty wide range, and it would make sense that some of the protein pairs mixed at < 0.5mg/mL would be much less likely to produce inter-links, even after heating. However, I don't see any such info for specific proteins, so it's hard to confirm/deny such hypotheses. Another possibility is that some fraction of proteins aggregate and precipitate out of solution due to heating before they have a chance to get cross-linked.

3) While it is stated that 256 proteins “were provided by Absea Biotechnology”, it seems a significant fraction of them might not have expressed, or at least they are not listed in the Supplementary Data 1 pdf. This would not be surprising in such a large-scale operation, but it would be helpful to include such details in the Methods somewhere, rather than keep the readers guessing. This would, in turn, help adjust expectations of the max # cross-linked protein pairs we should expect to see.

4) The details of how exactly the proteins were mixed needs to be included in supplementary materials with the paper, not just in PRIDE.

5) According to Suppl. Table 2, the authors for some reason have chosen to use KSTY reactive sites for Scout and XiSEARCH but not for other search engines. This seems like a very strange choice – this is clearly an available option for most if not all of them (MS Annika, XlinkX and MeroX were all compared with K vs KSTY in the Matzinger et al paper suppl.). This parameter may make a difference depending on the specifics of the search engine and/or sample prep conditions. If KSTY searches from Matzinger are not available or obtainable, it would make sense to just use K-K for Scout as well for a fair comparison in relevant figure/tables. This absolutely must be addressed.

6) What is the ANN implementation used? GNB? Did the authors write their own or rely on existing libraries? The number of input parameters used is pretty small so simpler discriminant approaches might have worked just as well, while also providing insight into parameter weights, rather than the ANN black box. This would be interesting to help figure out if the improved performance of Scout is primarily due to a large training set or the input scores, which don't appear to be particularly novel.

7) Authors mention that a portion of the dataset was used for training, and another, independent portion for testing, but it's not clear specifically which portions were used for what purpose and how big they are in terms of number of proteins/pairs. The search results in PRIDE appear to refer to 3rd and 4th batches, but it's not clear what those refer to. Are the search results for 1st and 2nd batches missing in PRIDE?

8) Line 147: "For intra-links, ... the most relevant information comes from the ResPair level, for which Scout shows the best overall performance (Supplementary Figure 1)."

I don't disagree with this statement, but I think it would be interesting to try to understand why Scout was outperformed (e.g. by MSAnnika) by up to ~50% on CSM level, and by ~20% on the protein level, but not on the ResPair level! Could this have something to do with how FDR is controlled, or the ratio of inter/intra links in the training dataset? It is likely Scout would be used by the community for small-scale datasets and single proteins as well, so it's important to understand how it performs in different circumstances and why. Again, it's not clear which portions of the dataset are used here. The CSM numbers don't appear to quite line up with the identifications_thirdbatch_fourthbatch_allsearchengines file in PRIDE.

9) In Figure 4C, Scout appears to outperform XlinkX by a wide margin on CSM level, and even wider margin on ResPairs level, but slightly under-performing (540 fasta) on the PPI level. This is a bit surprising and some discussion of possible causes is warranted. E.g., Scout vs xiSearch profile makes a lot more sense with gradually decreasing outperformance from CSM to PPI level.

10) I downloaded the latest release of Scout. The interface looks nice and is fairly intuitive, but when I tried running it with default params on one of the RAW files in PRIDE. The result was the following error:

```
ERROR - PerformSearch method: It's not possible to run the search.
at System.Linq.ThrowHelper.ThrowArgumentOutOfRangeException(ExceptionArgument argument)
at System.Linq.Enumerable.Chunk[TSource](IEnumerable<T> source, Int32 size)
at ScoutCore.PSMEngines.Tripper.Tripper.RunSpectraSearchParallel(List<T> allQueries, List<T> allPeptides)
at ScoutCore.PSMEngines.Tripper.Tripper.PerformSearch(String fastaFile, String rawFile)
Specified argument was out of the range of valid values. (Parameter 'size')
```

** For Nature Portfolio general information and news for authors, see <http://npg.nature.com/authors>.

Version 1:

Decision Letter:

4th Jan 2024

Dear Fan,

Happy New Year!

Thank you for your letter asking us to reconsider our decision on your Article, "Proteome-Scale Recombinant Standards and a Robust High-Speed Search Engine to Advance Cross-Linking MS-Based Interactomics". After careful consideration we have decided that we are willing to consider a revised version of your manuscript with the proposed revisions.

Regarding the two points you wanted our feedback on - the plan to include an analysis of the unpublished XL-MS dataset generated from intact HEK293 cells sounds good. Do you plan to validate any of the findings from this dataset? Regarding comparing stepped-HCD MS/MS and MS3 strategies on PPI identification that has previously been done by Matzinger et al. is not necessary, but please provide the relevant citation(s) and discussion.

* include a point-by-point response to our referees and to any editorial suggestions

* please underline/highlight any additions to the text or areas with other significant changes to facilitate review of the revised manuscript

* address the points listed described below to conform to our open science requirements

* ensure it complies with our general format requirements as set out in our guide to authors at www.nature.com/naturemethods

* resubmit all the necessary files electronically by using the link below to access your home page

Link Redacted

We hope to receive your revised paper within 8 weeks. If you cannot send it within this time, please let us know. In this event, we will still be happy to reconsider your paper at a later date so long as nothing similar has been accepted for publication at Nature Methods or published elsewhere.

OPEN SCIENCE REQUIREMENTS

REPORTING SUMMARY AND EDITORIAL POLICY CHECKLISTS

When revising your manuscript, please submit reporting summary and editorial policy checklists.

IMAGE INTEGRITY

DATA AVAILABILITY

CODE AVAILABILITY

Please include a "Code Availability" subsection in the Online Methods which details how your custom code is made available. Only in rare cases (where code is not central to the main conclusions of the paper) is the statement "available upon request" allowed (and reasons should be specified).

MATERIALS AVAILABILITY

As a condition of publication in Nature Methods, authors are required to make unique materials promptly available to others

without undue qualifications.

SUPPLEMENTARY PROTOCOL

To help facilitate reproducibility and uptake of your method, we ask you to prepare a step-by-step Supplementary Protocol for the method described in this paper. We [encourage authors to share their step-by-step experimental protocols](https://www.nature.com/nature-research/editorial-policies/reporting-standards#protocols) on a protocol sharing platform of their choice and report the protocol DOI in the reference list. Nature Portfolio's Protocol Exchange is a free-to-use and open resource for protocols; protocols deposited in Protocol Exchange are citable and can be linked from the published article. More details can found at www.nature.com/protocolexchange/about.

ORCID

Sincerely,
Arunima

Arunima Singh, Ph.D.
Senior Editor
Nature Methods

Version 2:

Decision Letter:

Our ref: NMETH-A54232B

19th Jun 2024

Dear Fan,

Thank you for submitting your revised manuscript "Proteome-Scale Recombinant Standards and a Robust High-Speed Search Engine to Advance Cross-Linking MS-Based Interactomics" (NMETH-A54232B). It has now been seen by the original referees and their comments are below. The reviewers find that the paper has improved in revision, and therefore we'll be happy in principle to publish it in Nature Methods, pending minor revisions to satisfy the referees' final requests and to comply with our editorial and formatting guidelines. As discussed, the code will also be made publicly available.

TRANSPARENT PEER REVIEW

Please note: we allow redactions to authors' rebuttal and reviewer comments in the interest of confidentiality. If you are concerned about the release of confidential data, please let us know specifically what information you would like to have

removed. Please note that we cannot incorporate redactions for any other reasons. Reviewer names will be published in the peer review files if the reviewer signed the comments to authors, or if reviewers explicitly agree to release their name. For more information, please refer to our [FAQ page](https://www.nature.com/documents/nr-transparent-peer-review.pdf).

ORCID

Sincerely,
Arunima

Arunima Singh, Ph.D.
Senior Editor
Nature Methods

Reviewer #1 (Remarks to the Author):

As already mentioned in the first review round, Scout seems an impressively fast and well working software, even further improved now, which will be of great value to the community. The authors carefully and comprehensively answered all reviewer questions and significantly improved the quality of their manuscript by adding a novel DSBSO based proteome-wide dataset. That said I would like to congratulate the authors to their excellent work and recommend publication of a revised version. Here are some minor comments:

1.
I appreciate the addition of Table R2 comparing used benchmarks to real datasets. Especially given the differences in dynamic range and ratio of linear peptides to crosslinked peptides. In that table the authors mention the Matzinger et al dataset based on synthetic peptides. To the best of the reviewer's knowledge Matzinger et al digested the synthetic peptides using trypsin and performed a reduction step to remove protecting parts and obtain tryptic peptides as would be present from a standard digest. Therefore, I would suggest something like adding "Synthetic peptides, final peptide obtained by protease digestion" (or similar) to the table to clarify the exact design (which is not as different as it seems in the current table). Regarding the question of PPI from the Matzinger et al. dataset I have the following thought: It's true that there are no real PPIs but from the perspective of the data analysis software there are PPIs since the used synthetic peptides correlate to 38 different proteins therefore yielding inter-protein crosslinks in the search. I would recommend adopting the text in Table R2 accordingly. In line I would recommend a similar rephrasing in the manuscript text, i.e. line 301 in the revised manuscript, could be rephrased to something like "Matzinger et al contains only limited PPI based information" (or similar).
2.
In Figure 2 I would recommend to add as possible input ".d" MS2 files as well, since this is supported by Scout.
3.
The newly added HEK-DSBSO dataset shows impressive coverage with ~2000 PPIs found. I am assuming that crosslinks were predominantly found from abundant protein complexes within the HEK cells. However, since the coverage is that excellent, I was wondering if the authors also found something less abundant in there? (Compare to Nouchikian et. al., <https://pubs.acs.org/doi/10.1021/acs.analchem.3c04682> who reported crosslinks can only be found at the top 20% most abundant proteins.) Where there crosslinks from all cellular compartments including the nucleus present?
4.
Furthermore, I was wondering how many cells/how many dishes were used in total to obtain the aforementioned results from HEK+DSBSO? I could only find the protein concentration of 10mg/mL but would be curious about the cell# needed for the described workflow involving the orthogonal bead based + SEC based enrichment. This would be of interest also for the community to estimate how much material is lost during enrichment.

Reviewer #2 (Remarks to the Author):

I am happy with the revision and recommend publication
Mikhail Savitski

Reviewer #3 (Remarks to the Author):

The revised manuscript is much improved and many of the crucial missing details have been filled in. I believe it is essentially ready for publication with a few fairly minor issues:

1) Because of the benchmarking nature of this work, the data is likely to be used by the community quite a bit, so it would be useful to understand a bit more about the raw files in PRIDE. While there's a Methods sub-section describing SCX, it's not actually clear where this was used. Were the synthetic benchmark samples fractionated with SCX? The number of raw files varies ~25-40 between different batches – is that because of different fraction pooling? Or were these not fractionated at all and each raw file is a result of pooling some number of mixed protein groups? Maybe an excel file with raw file metadata would be useful for people so that they know what they are searching.

2) Supplementary Notes 1-4 now provide quite a bit of detail on the implementation of Scout which might be useful to some readers. However, as the authors themselves explain in the rebuttal (36):

“The advantages of Scout arise from thoroughly optimizing details of the methodology of fragment match collection, ion selection in theoretical spectra, spectrum pre-processing ..., and ion intensity handling. These seemingly minor changes are crucial because minimal modifications of these aspects can significantly impact identification accuracy and the filtration of decoy hits.”

On the software side, the objective of an academic manuscript ideally shouldn't be just to act as an advertisement for software downloadable elsewhere, but to also try to convey what was learned during its development. Since the software is not open source, it's important to make sure that such details the authors consider crucial above are included in these supplementary notes.

3) Line 497 - “high-pH separated enriched phospho peptides”

Since there's no phospho-peptide analysis in this manuscript, I assume this to be copied/pasted from unrelated manuscript?

Version 3:

Decision Letter:

19th Sep 2024

Dear Fan,

I am pleased to inform you that your Article, "Proteome-Scale Recombinant Standards and a Robust High-Speed Search Engine to Advance Cross-Linking MS-Based Interactomics", has now been accepted for publication in Nature Methods. The received and accepted dates will be October 23, 2023 and September 19, 2024. This note is intended to let you know what to expect from us over the next month or so, and to let you know where to address any further questions.

Over the next few weeks, your paper will be copyedited to ensure that it conforms to Nature Methods style. Once your paper is typeset, you will receive an email with a link to choose the appropriate publishing options for your paper and our Author Services team will be in touch regarding any additional information that may be required. It is extremely important that you let us know now whether you will be difficult to contact over the next month. If this is the case, we ask that you send us the contact information (email, phone and fax) of someone who will be able to check the proofs and deal with any last-minute problems.

Please note that *Nature Methods* is a Transformative Journal (TJ). Authors may publish their research with us through the traditional subscription access route or make their paper immediately open access through payment of an article-processing charge (APC). Authors will not be required to make a final decision about access to their article until it has been accepted. [Find out more about Transformative Journals](https://www.springernature.com/gp/open-research/transformative-journals)

If you have posted a preprint on any preprint server, please ensure that the preprint details are updated with a publication

reference, including the DOI and a URL to the published version of the article on the journal website.

If you are active on Twitter/X, please e-mail me your and your coauthors' handles so that we may tag you when the paper is published.

Best regards,
Arunima

Arunima Singh, Ph.D.
Senior Editor
Nature Methods

** Visit the Springer Nature Editorial and Publishing website at http://editorial-jobs.springernature.com?utm_source=ejP_NMeth_email&utm_medium=ejP_NMeth_email&utm_campaign=ejp_Nmeth for more information about our career opportunities. If you have any questions please click [here](mailto:editorial.publishing.jobs@springernature.com).**

REVIEWER COMMENTS

General note: We have numbered all reviewer comments consecutively to make this document easier to navigate

Reviewer #1:

Remarks to the Author:

In their manuscript, the authors describe a mix of synthetic proteins as new benchmarking tool to validate FDR of cross-linking MS software at PPI level that is designed to mimic complex samples as whole cell cross-link experiments. It's therefore a nice complementary tool for benchmarking on PPI level compared to existing peptide-libraries that allow to validate FDR on residue pair level. The authors use the synthetic protein mix to develop their own novel XL-MS search engine, Scout. Scout delivers excellent results by means of identified XLs and PPIs while maintaining a low FDR as shown by usage of a published peptide library and their own protein mix. Full transparency is guaranteed as all raw files are made available via PRIDE and the code is made available on github.

With no doubt the new software, as well as the datasets created on the synthetic peptide mix will be of value for the community, especially as the software is free of charge, easy to use and seems to produce reliable results.

We thank the reviewer for taking the time to assess our submission, and for the constructive and overall positive feedback.

1. This reviewer is concerned though, if the current work is suitable for Nature Methods due to the lack of an application of the new method to a biological question demonstrating its advantage over existing approaches. Such an application could however be added to a revised version of this manuscript, even the reanalysis of existing XL studies finding new insights with Scout would demonstrate its applicability to a biological question.

We agree with the reviewer. As described below, we analyzed several biological XL-MS datasets using Scout and, for comparison, MSAnnika. We chose MSAnnika because this software (1) was among the best performing in our benchmarking analysis, (2) can complete complex searches against large databases on the desktop PCs available to us, (3) shows one of the most robust performances when increasing database size, and (4) does not depend on manual post-processing with heuristic score cut-offs. The other well-performing search engines in our benchmarking analysis did not meet all of these criteria; specifically, xi did not comply with point 2 and XlinkX did not comply with points 3+4.

We first performed an entrapment experiment, searching a published XL-MS dataset from Human Mitochondria cross-linked with the enrichable, and MS-cleavable Azide-A-DSBSO cross-linker (Zhu et al., Nat Commun 2024, 10.1038/s41467-024-47569-x) against a database comprising the proteomes of human mitochondria and E.coli. In this setup, all cross-links between human proteins are considered true and cross-links involving E.coli proteins are considered false. Entrapment does not account for incorrect human-human cross-links, meaning that it is less sensitive than the FDR estimation based on our standard datasets and

will likely underestimate the actual FDR. Nonetheless, it gives an indication how Scout's sensitivity and specificity compare to other search engines for real-world data.

The results of the entrapment search in Figure R 1A largely confirm the trends observed in the benchmarking with our standard dataset: Scout identifies the most CSM and ResPair hits, while providing fewer – but more stringently FDR-controlled – identifications on PPI level. For the reasons detailed above, the FDR differences are much less pronounced than in the analysis of our standard datasets. However, the effectiveness of Scout's PPI-FDR is demonstrated by the fact that Scout is the only search engine that completely removes forbidden PPIs. To illustrate the effect of the PPI-FDR filter, we also report Scout PPI identifications aggregated from ResPair level (i.e. the same approach used for MSAnnika). The aggregated results are highly similar to the other search engines, showing that stringent PPI-FDR control is only possible with a dedicated filter. This filter will inevitably impact PPI identification numbers, as also demonstrated in our previous comparisons to xi, which is the only other search engine with FDR control on PPI level and reports lower PPI numbers than Scout (Figures 4D and 6C).

Next, we applied Scout and MSAnnika to unpublished data obtained by cross-linking intact HEK293T cells with Azide-A-DSBSO. To assess and validate the biological information content of the results, we compared

- the identified PPIs to the STRING database and the Negatome database of non-interacting proteins (Blohm et al., NAR 2014, 10.1093/nar/gkt1079)
- ResPair cross-links to AlphaFold (AF)-Multimer models of the PPIs identified by the two search engines

The STRING analysis (Figure R 1B) confirms our previous observation that, compared to MSAnnika, Scout's PPI-FDR filter reduces PPI identification numbers. At the same time, the PPI-FDR filter slightly increases the fraction of medium-to-high-confidence PPIs (79% for PPI-FDR controlled Scout vs 75% for MSAnnika and 76% for Scout with PPIs aggregated from ResPair identifications). PPI-FDR-controlled Scout also identifies the fewest hits from the Negatome database, suggesting that the PPI-FDR contributes to identification specificity.

Since we repeatedly observed that Scout shows a steeper numerical drop-off from ResPair to PPI level than the search engines without dedicated PPI-FDR filter, we also looked into the relationship between ResPair and PPI level. For the HEK dataset, we find that the surplus of MSAnnika PPI identifications mainly stems from PPIs supported by a single ResPair, whereas Scout has an edge over MSAnnika in identifying PPIs supported by >6 ResPairs (Figure R 1C). The difference between Scout and MSAnnika in the single ResPair category disappears when aggregating the PPI data from Scout's ResPair identifications, indicating that Scout's PPI-FDR filter mainly removes weakly supported PPIs that are more likely to represent irreproducible "one hit wonders".

Our comparison of ResPair inter-links to AF-Multimer models shows that Scout provides the highest number of ResPair inter-links that can be mapped onto AF-Multimer models (Figure R 1D). The structural accuracy of Scout MSAnnika inter-links is similar, with a slight advantage for MSAnnika (77.8% within-distance inter-links vs 75.4% for Scout). However, both software tools perform essentially identical across the full range of AF-Multimer model confidence scores (Figure R 1E).

Overall, Scout's greater identification depth on CSM and ResPair level can provide additional information for structural biology applications. At the same time, Scout's stringent PPI-FDR filter, although reducing identification sensitivity, helps remove questionable PPIs, thereby increasing the chance that the remaining hits are readily actionable for functional follow-up studies.

Figure R 1. Application of Scout and MSAnnika to biological proteome-wide XL-MS datasets.

- (A) Entrapment database search on an Azide-A-DSBSO cross-linked human mitochondria dataset (Zhu et al., 2024). The data was searched with 2000 random human mitochondria proteins sampled from a linear peptide search on the XL-MS data, supplemented with 2000 random *E. coli* BL21 protein sequences. Interspecies cross-links and *E. coli* cross-links were considered false. The percentage states the resulting empirical false positive rate. PPI-level results for Scout were either determined using the PPI-FDR filter (Scout) or by aggregation of ResPairs to unique protein pairs (Scout*). The second approach was also used for MSAnnika.
- (B) Evaluation of PPIs identified from a HEK cell Azide-A-DSBSO XL-MS dataset on protein-protein interaction databases. Brown, light blue and dark blue correspond to different STRING confidence score ranges. Yellow represents identifications that could not be found in STRING or that are considered impossible because they match to the Negatome database. PPI-level results for Scout were either determined using the PPI-FDR filter (Scout) or by aggregation of ResPairs to unique protein pairs (Scout*). The second approach was also used for MSAnnika.
- (C) Number of ResPair inter-links per PPIs identified with MSAnnika and PPI-FDR controlled Scout on the Azide-A-DSBSO HEK dataset.

- (D) $C\alpha-C\alpha$ distances of ResPair inter-links identified by Scout and MSAnnika, when mapped on AF-Multimer models of their identified PPIs. For each PPI, the model with the highest cross-link satisfaction was used for analysis. The plot considers all inter-links between residues with a pLDDT score above 50 (indicating an ordered protein region) that can be mapped on AF-Multimer models with a model confidence of at least 0.5.
- (E) Spread of inter-link $C\alpha-C\alpha$ distances for different ranges of AF-Multimer model confidence. Only inter-links between residues with a pLDDT score above 50.

To complement our HEK cell analysis with another intact-cell XL-MS dataset obtained with a different cross-linker, we turned to a recent study that developed the glycosidic bond-based MS-cleavable cross-linker trehalose disuccinimidyl ester (TDS) and applied it to intact HeLa cells (Chen et al., *Angew Chem* 2023, doi.org/10.1002/anie.202212860). The authors reported identification of 842 PPIs (810 after removing duplicates), but their Methods section suggests that this result was obtained using XlinkX PD 2.2 without post-processing (i.e. no XlinkX score or delta score cut-offs). This approach may cause a sizable overestimation of PPIs, considering that our benchmarking results show a massive inflation with wrong PPIs for XlinkX searches against large protein databases even when using the newer XlinkX PD 2.5 (with improved FDR control) and applying a static XlinkX score cut-off (Figure 4A). Confirming this expectation, a re-analysis of the TDS-HeLa data using XlinkX PD 2.5 with dynamic score filtering (i.e. the optimal XlinkX setup according to our benchmarking analysis) yielded less than 100 PPIs; Scout and MSAnnika achieved similarly low identification numbers. These drastic differences illustrate the risk of using software tools that rely on heuristic score cut-offs instead of an inbuilt, robust FDR control. The very limited PPI-level depth obtained with appropriately controlled search engine settings may be a consequence of TDS being a non-enrichable cross-linker and insufficient fractionation of the HeLa sample (only high pH fractionation), as we and others have shown that cross-linker enrichability and sample fractionation by SEC/SCX are key factors to achieve deep interactome coverage in intact human cells (e.g. Wheat et al., *PNAS* 2021, 10.1073/pnas.2023360118 and Jiang et al., *Angew Chem* 2022, 10.1002/anie.202113937). Although the low coverage makes this dataset unsuitable for software comparisons on PPI level, we still analyzed the CSM and ResPair level output of Scout to demonstrate that the software is able to process TDS crosslinking data (Figure R 2). However, in the absence of a meaningful PPI-level comparison we would prefer to omit these data from the manuscript.

Figure R 2. Scout CSM and ResPair inter-link identifications from intact HEK cells cross-linked with TDS (Chen et al., *Angew Chem* 2023, doi.org/10.1002/anie.202212860). Search was performed using standard parameters (see Methods and Supplementary Table 2) and the uniprot *H.sapiens* reference proteome (downloaded in May 2020).

Action taken: We have added Figure R 1 as a new main text figure. The data are described in the final Results section and in Supplementary Note 5.

Please find below my comments and questions to the current manuscript:

Scout: In the hands of this reviewer Scout worked out of the box. It comes with an intuitive to use GUI that allows to adopt search parameters, define custom linker reagents, enzymes, or modifications. It works only with MS cleavable linkers, but for the aimed usage of proteome wide searches it makes sense to focus on such linker types. Furthermore, Scout offers export functionalities for post processing. Most importantly, Scout is indeed very fast and needs less resources compared to other commonly used software and gives a reliable FDR which makes it an excellent choice for proteome wide analyses using MS cleavable linkers! For direkt software-internal visualization, Scout comes with a very nice spectrum viewer and outputs some helpful plots for statistics and QC.

Here some suggestions for improvement:

2. A direct export for usage of XlinkCyNet, a data visualization tool from the author's lab, is possible. Direct support for more visualization tools such as xiView would even improve the usability for a broad range of users here.

We fully agree.

Action taken: We have implemented an exporting method to generate mzIdentML 1.2.0 file, which is compatible with xiView. This information has been added to the Methods section (sub-section "Scout").

3. • While all works fine with Thermo raw-files and this reviewer obtains similar numbers as shown in the manuscript with some sample files, the analysis of Bruker (.d) files seems problematic. When loading a ".d" file, Scout crashes in my hands and when loading the converted mgf instead, it gives only ~1/5 of the crosslink IDs compared to MaxLynx. Did the authors test to analyze Bruker files themselves and can report their results in an updated manuscript? This would certainly further improve its quality and would broaden the potential user-group.

Thanks for pointing this out Scout previously has had problems to process ".d" files in certain environments (e.g. when run on virtual machines). The new version Scout 1.5.0 (<https://github.com/diogobor/Scout/releases/tag/1.5.0>) runs more stably on virtual machines as well (when complying with the hardware requirements specified on the Scout GitHub page: <https://github.com/diogobor/Scout>), but please note that we recommend running Scout on Windows platforms (see software requirements on the Scout GitHub page).

We have used published ".d" files (Ihling et al., Anal Chem 2021, 10.1021/acs.analchem.1c01317) to confirm that Scout is able to process this file type. Similarly, Scout will accept ".mgf" files as the reviewer stated. That being said, the structure of ".d" files and the derived ".mgf" files may vary depending on the type/settings of the instrument used to record the MS data (e.g. whether TIMS separation is performed) and the MGF converter. We were unable to comprehensively assess the effects of these variations based on published datasets and we did not have access to Bruker instruments to record our own data. Assessing the number of cross-link identifications from Bruker-derived data is further

complicated by the fact that our XL-MS ground-truth data as well as all previously published XL-MS (pseudo-) ground-truth datasets (Beveridge et al., Nat Commun 2020, 10.1038/s41467-020-14608-2, Lenz et al., Nat Commun 2021, 10.1038/s41467-021-23666-z, Matzinger et al., Nat Commun 2022, 10.1038/s41467-022-31701-w) have been acquired on Thermo instruments. This means that we cannot confidently evaluate to which extent the total identification numbers are inflated by false-positive identifications, which precludes a fair software comparison. The importance of such a benchmark is illustrated in Figure 3, where MeroX and MS Annika yield substantially more PPIs than Scout, but these identifications are intermingled with a large fraction of false-positive hits. It is possible that a similar phenomenon caused the difference in total identification numbers that the reviewer observed.

In addition, using Bruker instruments for proteome-wide XL-MS has not been broadly explored yet (we only know of proof-of-concept experiments by Steigenberger, MCP 2020, 10.1074/mcp.RA120.002094, and Ihling et al, Anal Chem 2021, 10.1021/acs.analchem.1c01317). This indicates that using Bruker instruments for comprehensive PPI mapping by XL-MS in biological applications will require additional collaborative development efforts by researchers and the instrument vendor.

Action taken: We now mention that Scout has only been benchmarked on data derived from Thermo instruments because XL-MS ground-truth standards for Bruker instruments are lacking and XL-MS with Bruker instruments is not yet being broadly used for proteome-wide XL-MS (see Supplementary Note 6 and Discussion). We also add more information on the use of “.d” and “.mgf” files to the Methods section (subsection “Scout”).

4. Synthetic protein mix:
The authors created a mix of 32 groups of 8 proteins each resulting in close to 900 true positive PPI hits. This allows to experimentally validate FDR on PPI level, while FDR on residue pair or spectrum match level can only be estimated. In this study all intra-protein crosslinks are considered as true positive, which however is only an assumption and not a real ground truth. This is important to mention also in the discussion of this study.

The reviewer highlights an important point that deserves further discussion. Although we cannot achieve perfect control over FDR at the residue pair or spectrum match level, our experimental design makes it possible to obtain highly accurate FDR estimates also at these levels:

- 1) Our strategy of inducing protein interactions through heat aggregation, as opposed to fixed orientations, facilitates a wide array of conformations and binding interfaces. This substantially increases the probability that all possible lysine-lysine contacts will be stochastically sampled.*
- 2) Previous studies (e.g., Giese et al., Nat Commun 2021, 10.1038/s41467-021-23666-z, already cited in our first submission) that false-positive intra-links are extremely rare.*
- 3) We performed mathematical simulations to estimate the expected fraction of false identifications on ResPair level that would be annotated as true-positives according to our mixing scheme (Figure R 3). The simulations are based on simple assumptions and controlled parameters, adopting an approach we described recently (see Related Manuscript included in this submission). The results show that fraction of false intra-links is always below 0.03%, even when searching small databases and*

allowing an empirical FDR of 10%. The fraction of false inter-links expected to occur within one interaction group is higher but remains at 0.12% (540-entry database) and 0.03% (4000-entry database) when setting a 1% empirical FDR cut-off.

Figure R 3. Expected fraction of false in-group ResPair cross-links in the XL-MS datasets derived from our recombinant analytical standard. For this simulation, we considered the 2 database sizes used for our comparison of XL-MS search engines and 3 FDR cut-offs (determined using target-decoy competition in Scout). The bars show the percentage of false ResPair intra-links and inter-links that are expected within one interaction group, i.e. cross-links that would be wrongly annotated when defining true-positive hits based on our mixing scheme. Shown are the average +/- SD from 10 simulations with identical input parameters.

We agree that our analytical standards qualify as a bona fide ground-truth only on PPI level; however, based on the above reasons, we can also confidently assume that false intra-link and in-group inter-links are extremely rare.

Action taken: In the first Results sub-section (“Constructing an XL-MS standard mimicking complex biological samples”), we have elaborated on the above points and clarified that our analytical standard serves as a ground-truth on PPI level and an "experimental benchmark for FDR validation" on CSM and ResPair level. This terminology more accurately reflects the experimental design and the usability of the dataset. We also added Figure R 3 as Supplementary Figure 1 and provide an explanation of the simulation approach in the Methods section.

- To build the library both proteins and protein fragments were used. I was wondering if there are cases of crosslinked residue pairs to sequence parts of proteins not existing (meaning that they might be counted correct but since only a protein fragment is present in the sample the software might output a crosslink to a part of the protein that is not part of the synthesized fragment). Have you checked on that and have you included full protein sequences into the FASTA files of your searches or only the existing fragments?

The FASTA files with all protein sequences are included in our PRIDE submission. They include the full uniprot sequences of all proteins. We considered identified cross-linked peptides that correspond to protein regions not included in our recombinant protein constructs as false-positive identifications but, in fact, we did not identify any cross-links from this category. We apologize for not mentioning this in the original submission.

Action taken: We included information on the FASTA file composition and treatment of peptides not present in the protein constructs it in the revised Methods section.

6. • What's the share of homologous peptides across the protein groups used for crosslinking?

Considering the full analytical standard (batches 1-4), the pipetted proteins give rise to 23,895 peptides within the constraints of Scout's search settings (3 missed cleavages, min. pep length 6, min. pep mass 500 Da, max. pep mass 6000 Da). Shared peptides occur frequently within the His-tag used for protein purification (80 in total). Outside of the His-tag, two peptides were shared between 3 proteins, and 397 peptides (1.66%) were shared between two proteins. As already mentioned in the Results sub-section "Benchmarking Scout on independent XL-MS standards", for cross-links involving shared peptides all possible annotations have been considered true on CSM and ResPair level.

Action taken: We have added this information to the description of our analytical standard (first sub-section of the Results).

7. Further comments:
Supplementary Table 2: The search parameters of the benchmarked software tools are not equal, most importantly most tools were set to search links from K to K but Scout is allowed to search for links to K, S, T, and Y. This seems not fair for comparison as Scout is "allowed" to find way more combinations for unique residue pairs. Please either explain why parameters were chosen like that or correct for harmonized parameters across all tested search engines.

We apologize for not making this very important point clearer in our initial submission. For all comparisons to MeroX, MaxLynx, MS Annika and XlinkX PD, Scout was set to only search for K-to-K links.

K,S,T, and Y as cross-linking sites were only considered for the comparison of Scout vs xiSEARCH/xiFDR. The reason for this deviation was that xiSEARCH/xiFDR analyses were not feasible using the standardized settings that we applied for the other software tools (discussed on page 7, 2nd paragraph, lines 184ff of our original submission). To still allow comparing Scout to the best-possible performance of xiSEARCH/xiFDR, we asked one of the xi co-developers (Francis O'Reilly, see Acknowledgements) for guidance to optimize the xi software parameters (summarized on page 14, 3rd paragraph, lines 387ff of our original submission). Following their recommendations, we allowed K, S, T, and Y as cross-linking sites, as this is the default setting of xi. The xi searches shown in this manuscript had to be executed on an external computer cluster, since searches against our small 450-protein database would not run to completion on the standard PCs that are available at our institute (Matzinger et al. encountered similar issues in <https://doi.org/10.1038/s41467-022-31701-w>). Since these challenges restrict our ability to re-do the xi analyses in-house, we decided to allow KSTY specificity in Scout for this specific comparison.

Action taken: We have clarified in Supplementary Table 2 that KSTY specificity was only selected for the Scout-vs-xiSEARCH/xiFDR comparison and that Scout was limited to K specificity for all other comparisons. We also highlighted this in our description of the Scout-vs-xi comparison in the main text.

8. In addition, the number of allowed miscleavages is lowered to 2 for Xi while its at 3 for all others, why? Was this done to reduce needed search space (as the authors had to deal with very long runtimes for Xi)?

Setting two miscleavages for xi was part of the developer recommendations to enable completion of the xi searches. We agree with the reviewer that this will help reduce the search space.

9. • Figure3/4: On PPI level Scout delivers excellent FDR control, however as already mentioned by the authors themselves, a direct comparison is not possible as Annika and XlinkX do not report PPIs, hence do not have FDR control on that level.

We fully agree. However, many users of these tools have utilized the output of these search engines to characterize PPIs. Therefore, we believe it is important to characterize their performance on PPI level as fairly as possible, and make the users aware of the consequences of taking these PPI results without PPI-level FDR control. To do so, we have aggregated the ResPair level results to PPI level. We highlighted this choice in the figures (grey boxes with dotted frames in Figures 3 and 4), and, as the reviewer mentioned, had already explained it in the Results and Methods sections. Overall, we believe that PPI-level FDR control is one of the most important advantages Scout offers over MS Annika and XlinkX.

10. • The comparison to Xi makes sense and Scout reports more PPIs at lowered FDR (Figure 4 panel D). However, I was wondering about the following:
- Why the processing time of Xi is not shown in Figure4 panel E (but only in a supplementary figure). This would highlight the main advantage of Scout over Xi?

All analyses shown in Figure 4E were done with the full benchmarking dataset. However, for xiSEARCH/xiFDR analyses, we had to reduce the benchmarking dataset to 4 selected RAW files (as explained in the legend of Supplementary Fig. 2) because otherwise xi searches would not run to completion on the computational setup used for this speed benchmark. Therefore, the xi processing times cannot be directly compared to the processing times of the other tools and it would be misleading to include the xi data in Figure 4E.

Action taken: We have clarified in the legend of Figure 4E why xi is not shown in this plot and mention that the xi data can be found in Supplementary Fig. 2.

11. • Furthermore, I didn't fully understand why the ID numbers differ here for Scout:
- o Figure 3/540 protein FASTA: ~200 PPI at 1% empirical FDR
 - o Figure 4D/540 protein FASTA: ~210 PPI at 1.9% empirical FDR
 - o Supplemental Figure 2A/540 protein FASTA: ~ 225 PPI at 2.7% empirical FDR
 - And for Xi:
 - o Supplemental Figure 2A/540 protein FASTA: ~200 PPI at 11.5% empirical FDR
 - o Figure 4D/540 protein FASTA: ~180 PPI at 3.2% empirical FDR
- This reviewer suggests to re-design Figure 4 to enhance clarity for the reader.

The differences between Figure 4D and Supplementary Fig. 2A are due to different software-defined FDR cut-offs (1% in Figure 4D, 5% in Supplementary Figure 2A). This was already mentioned in the main text.

The differences between Figure 3 and Figure 4D are mainly caused by the fact that K,S,T, and Y were considered as cross-linking sites for the Scout-vs-xi comparison, whereas only K was considered as cross-linking site for the comparisons of Scout against the other software tools (see also our response to comment 7). KSTY specificity was already mentioned in the legend of Figure 4D, and we apologize for not highlighting in the legends of Figure 3 and 4A-C that K-only specificity was used.

Action taken: We have specified in the legends of Figures 3 and 4A-C which cross-linking site specificity was set.

12. • The addition of a list that defines which protein in which group in order for others to validate/use the dataset would improve broad application of this “protein library” as a tool for the community.

Action taken: We have included this information in Supplementary Data 1.

Reviewer #2:

Remarks to the Author:

The present work provides a very important and highly useful dataset + software for the community. The ability to estimate PPI error is unique and the ability to control it explicitly is also helpful (and not standard across different softwares). Software is impressively fast, and it is clear that care has been taken to ensure accuracy of the identifications at all levels of cross-link identification. I am overall very enthusiastic about the paper which will have a substantial impact on the crosslinking field and would suggest publication after the comments below are addressed.

We very much appreciate this positive assessment of our work.

13. Line 43 - Reference 6 is probably not the most appropriate citation for this statement, as it is only mentioned briefly and does not contribute substantially to their quality control filtering of the dataset. Reference PMID:30649854 explicitly (and successfully) applied structural mapping to segregate correct and incorrect crosslink spectral matches. Other papers also do this as it is the convention in the field. I suggest to either (1) cite more large-scale XL-MS papers that perform some structural mapping as benchmarking, or (2) reword the statement to say something to the effect of: One way that XL-MS search outputs can be assessed is by....

We appreciate this suggestion.

Action taken: Instead of reference 6 we now cite the suggested reference (Giese et al., *Anal Chem* 2019, 10.1021/acs.analchem.8b04037) and other large-scale XL-MS papers (Mintseris et al, *PNAS* 2020, 10.1073/pnas.1902931116 as well as Mendes et al., *Mol Syst Biol* 2019, which was already cited in a different place in our manuscript).

14. Line 81 - suggest explicitly mentioning how cross-link production was maximised in the main text here (20 min incubation at 50C to cause aggregation before crosslinking). On a related note, it would be useful to mention how many of the potential 896 PPIs you were able to detect crosslinks for, using your aggregation approach. Were there any known bona fide PPIs present in the mix?

We agree that the experimental conditions need to be clearly mentioned in the main text. We also agree that the PPI coverage warrants additional discussion. The number of identified PPIs depends on multiple search parameters including the choice of search engine, size of the protein sequence database and the selected FDR cut-off. The maximum of 896 PPIs refers to the full analytical standard. Since we only used half of the data for software benchmarking, the tools could have detected up to 448 PPIs. As shown in Figures 3 and 4d, Scout identifies 195 PPIs or 209 PPIs at 1-2% empirical FDR, depending on whether only K or K, S, T, Y were allowed as cross-linking sites. This corresponds to a PPI coverage of 43.5% and 46.6%, respectively. This incomplete PPI detection was an expected result for us, because we intentionally designed the standard sample such that the structure of the resulting XL-MS data will closely resemble a complex proteome-wide XL-MS dataset where only part of the PPIs can be detected due to sample complexity. As such, the obtained PPI detection rate illustrates an asset of our experimental strategy.

As the reviewer was curious about the overall PPI coverage for the full analytical standard, we performed an additional Scout search using the small 540-protein database and a more relaxed naive PPI-level FDR cut-off of 5%. This search yielded 488 true PPIs (54.4% of theoretical maximum), confirming that PPI coverage is incomplete and depends on the specific search parameters. Since one of the four datasets (8 out of 32 interaction groups) was only used for Scout development and not for search engine testing, we opted to omit the analysis of the full analytical standard from the manuscript to avoid confusion and instead discuss the incomplete PPI coverage based on those datasets that were used for search engine benchmarking.

We did not consider bona fide PPIs in the mixing scheme, because we cannot guarantee that every bona fide PPI would be formed in our *in vitro* setup using recombinant proteins. We chose the heat aggregation protocol because it allows us to reliably induce contacts between the two proteins mixed in the same tube, independent of their physiochemical and structural properties. This choice was based on small-scale test experiments, in which heat treatment resulted in inter-link formation for 5 out of 5 protein pairs (Figure R 4).

Figure R 4. Map of intra-links (pink) and inter-links (cyan) for 5 randomly selected protein pairs used to test the suitability of heat treatment to induce cross-links.

Action taken: The parameters and rationale underlying the heat aggregation method are now described in the first sub-section of the Results. The incomplete PPI coverage is now discussed in the Results section when we describe the results of Figure 3.

15. Line 89 - ref 31267744 should also be cited. Also, the text should be tweaked - the fact that intra-protein XLs are mostly always true-positives is only really true for large scale XL-MS studies. Smaller databases (such as those used in small scale studies) matched to many cross-linked spectra may lead to a higher proportion of false-positive intralinks, as identifications are forced to match those proteins.

Action taken: We now cite the suggested reference (Keller et al., JPR 2019, 10.1021/acs.jproteome.9b00189) and clarify in the first sub-section of the Results that our assumption regarding intra-links is better applicable to large-scale XL-MS studies (i.e. the type of XL-MS studies for which our standard is intended).

16. Line 107 - How were these CSMs identified? Which program/s? It is not clear after reading both the main text and the methods section, and I was not able to identify which dataset this was in the PRIDE submission. Overall, I think there needs to be more statistics describing the results and composition of this synthetic training/evaluation dataset better.

We apologize for these ambiguities.

The raw files used to optimize Scout are labelled as “first batch” in the PRIDE submission. The 256 proteins were randomly put into 32 interaction groups to generate our analytical standards. The interaction groups were randomly allocated to 4 batches with 8 interaction groups each. Of these 4 batches, batch 1 was used for guiding the development of Scout. We realize now that CSMs are not well suited to estimate the comprehensiveness of an XL-MS dataset, because their count will always depend on the search parameters. We think it makes more sense to report the number of MS2 scans as this metric is independent of any analysis software. Our batch 1 dataset contains 1,409,900 MS2 spectra, which is in the same range as our HEK cell XL-MS dataset obtained with the Azide-A-DSBSO cross-linker (1,150,447 MS2 spectra). Thus, in term of the size of the MS2 spectra, batch 1 is comparable to a real-world proteome-wide XL-MS dataset.

Action taken: In the second Results sub-section (“Using the XL-MS standard to develop an artificial neural network-based cross-link search engine”), we changed the reported metrics to MS2-level. At the end of the first Result sub-section (“Constructing an XL-MS standard mimicking complex biological samples”), we added a paragraph to describe more clearly which parts of our standard were used for Scout development and software benchmarking, respectively. We have also added additional columns to Supplementary Data 1 that summarize the composition of our batches and interaction groups.

17. Line 107 - In line 89, you state that most erroneous IDs in XL-MS searches stem from interlink cross-links (as bad peptide matches are unlikely to both map to the same protein by chance in a large database). Given you want to model the error contributing to this high rate of inter-link false-positives by generating a synthetic dataset with PPIs, it would be important to explicitly report here how many CSMs you had for inter vs intra. I would guess that even with the 50C aggregation step, a large proportion of CSMs from your synthetic dataset would be intra-links. If the aggregation step was able to substantially increase the proportion on inter-link peptides and hence enrich your synthetic dataset with these species, this is worth emphasizing in the main text as a significant advantage of your dataset over others already published.

The reviewer is correct that the identifiable CSMs in our dataset correspond to intra-links. In the original submission, we reported these numbers in Figure 3 (inter-links) and Supplementary Fig. 1 (intra-links) but we did not explicitly mention them in the main text. Depending on the search engine, we detected 8,209 – 58,986 CSM-level intra-links at 0–24.86% empirical FDR and 6,273–9,530 ResPair-level intra-links at 0–30.92% empirical FDR. At the same time, we detected up to 1,781 CSM-level inter-links at 0.78% empirical FDR and up to 1,191 ResPair-level inter-links at 1% empirical FDR. For Scout, the fraction of inter-links was 4.75% at CSM-level and 11.42% at ResPair-level. This ratio is well in line with some of our previously published proteome-wide XL-MS data (see Table R 1 below), providing additional evidence that our analytical standard resembles many aspects of a ‘real-word’ XL-MS dataset (see also Table R 2 in our response to Reviewer#3). Table R 1 also shows that the percentage of inter-links varies widely, which is expected as it strongly depends on various parameters including sample type (e.g., if the proteins are in solution or in a tightly packed environment), cross-linker, MS acquisition strategies, cross-link search engine etc. In addition to resembling ‘real-word’ XL-MS data, our main goal when designing this analytical standard was to have a sufficiently large fraction of inter-links to enable Scout development and software benchmarking. The data presented in this manuscript provide evidence that this has been achieved.

Sample	% ResPair inter-links	Cross-linker	MS strategy	Reference
Intact mitochondria	60%	DSSO	CID-MS2-MS3-ETD-MS2, 2% FDR on CSM level (inter- and intra-links merged)	Liu et al., Mol Cell Proteom, 2018
HeLa cell lysate	11%	DSSO	CID-ETD-MS2, 1% FDR on CSM level (inter- and intra-links merged)	Liu et al., Nat Methods, 2015
HEK293T cell lysate	3%	tBu-PhoX	FAIMS-HCD-MS2, 1% FDR on CSM level (inter- and	Jiang et al., Angew Chem, 2022

			intra-links separated)	
Intact HEK293T cells	5.5%	tBu-PhoX	FAIMS-HCD-MS2, 1% FDR on CSM level (inter- and intra-links separated)	Jiang et al., Angew Chem, 2022
Recombinant XL-MS standard	11%	DSSO	Stepped-HCD-MS2, 1% FDR on ResPair level (inter- and intra-links separated)	This study

Table R 1. Fraction of inter-links detected in our recombinant standard and published proteome-wide XL-MS datasets.

Action taken: As mentioned in the previous response, we have removed the CSM metric, to which the reviewer is referring in their comment, and are now reporting MS2-level complexity. In the context of our software comparison, the number of identified intra- and inter-links is shown in Figure 3 and Supplementary Figure 1, which we now explicitly mention in the corresponding paragraph in the Results section.

18. Line 131 - it would be clearer to mention the number of “standard” datasets used, which is, if I understood correctly, “... benchmarked with the remaining 24 protein standard groups.”.

We apologize for the unclear phrasing. The usage of the interaction groups is described in detail in our response to this reviewer’s comment 16. We have only used 16 interaction groups (“third batch” and “fourth batch” in the PRIDE submission) for the software benchmarking. The second batch was used for the initial optimization of our LC-MS acquisition method and was not included in the software benchmarking (see our response to Reviewer#3’s comment 35 for more details). Since LC-MS method optimization is not of central importance to this study, we did not discuss it in the paper. However, we now performed additional method comparisons using batch 2 (stepped HCD-MS2 vs MS2-MS3 acquisition strategies) and we will include this information in a Supplementary Note (see our response to this reviewer’s comment 23).

Action taken: In the second Results sub-section (“Using the XL-MS standard to develop an artificial neural network-based cross-link search engine”), we added a paragraph to describe more clearly which parts of our standard were used for Scout development and software benchmarking, respectively. We also updated the sentence referred to in this comment, now explicitly stating which datasets were used for benchmarking.

19. In Figure 3, and related text, the terminology aggregated is used to describe the collapsing of error across redundant levels of XL-MS error (CSM to ResPairs to PPIs). Not necessary but, for clarity's sake and to help the reader, I would suggest a different terminology to clarify that these programs do not intrinsically control the FDR to these levels (hence the problem, and on the other hand, a key benefit of your program). For example, "manually collapsed", or "post-hoc aggregated", etc.

In the initial submission, this terminology was defined in the Results section as follows: "If a software tool did not provide naïve FDRs on all identification levels (CSM, ResPair, PPI), we took the naïve FDR-filtered identifications from the next lowest level and aggregated them to the higher level without any post-processing filters and score cut-offs (referred to as "aggregated results")."

Action taken: We have kept this description and, to further increase clarity, changed "aggregated" to "post-hoc aggregated" as suggested. In addition, we have changed "naïve FDR" to "software-defined FDR" to distinguish the FDR defined by the search engines more clearly from the FDR calculated from our standard datasets (which is called "empirical FDR").

20. Line 171 - The yeast cross-linking study used to define "default" XlinkX settings used in Figure 4A suggested a two-score in search cut-off to control PPI FDR in XlinkX - not just a minimum of XlinkX score 60 as used in this study, but also a minimum delta XlinkX score of 10 (Figure 4H of PMID:31851481). I would suggest to report either the results (1) from settings as optimised and reported in that study (minimum XlinkX score of 60, and delta score of 10), or (2) as from the default Proteome Discoverer 2.5 XlinkX settings with just 1% FDR settings in search, or (3) consider omitting and use just "dynamic" and empirically filtered XlinkX results (Figures 4B and C). The yeast study used an earlier version of Proteome Discoverer and hence version of XlinkX (which updates alongside with each Proteome Discoverer update and where each update performs different FDR control strategies). Therefore, I would suggest reporting either of the last two options would be most appropriate.

We thank the reviewer for these suggestions. Choosing the most suitable settings for XlinkX PD was not a trivial task and we are sorry that we did not explain our approach and the underlying rationale in sufficient detail. For the XlinkX PD search, we have used the default settings (minimum score 0, minimum delta 4, FDR 1% on CSM and Crosslink levels) and then performed post-search filtering. We only imposed an XlinkX score cut-off of 60 for three reasons:

- I) In our hands, including a delta score of 10 did not affect the results.*
- II) For the sake of clarity, we wanted to avoid introducing yet another scoring function.*
- III) Previous small-scale benchmarking studies (Beveridge, Nat Commun 2020, 10.1038/s41467-020-14608-2, and Matzinger et al., Nat Commun 2022, 10.1038/s41467-022-31701-w) did not use a delta score cut-off and recommended an even lower XlinkX score cut-off (41-45) for DSSO data.*
- IV) A static XlinkX score is the most intuitive cut-off and best comparable to the dynamic score filtering, which is solely based on the XlinkX score.*

We chose to report two versions of results (i.e., using a static score cut-off and a dynamic score filtering) because, on the one hand, most XlinkX PD users set static score cut-offs but,

on the other hand, the dynamic score filter is recommended by the software vendor as alluded to at <https://assets.thermofisher.com/TFS-Assets/CMD/Reference-Materials/pp-001448-ov-proteome-discoverer-presentation-software-application-pp001448-na-en.pdf> and in a recent publication applying XlinkX PD for disulfide bridge mapping (Heissel et al., MCP 2024, 10.1016/j.mcpro.2024.100759).

Since XlinkX PD users are completely free to set their preferred score cut-off, we included the result from both options to raise awareness of the impact that the XlinkX PD score cut-off setting will have on their results.

Action taken: In the Methods (sub-section “Data analysis”), we expanded our description of the XlinkX search parameters (now also mentioning the XlinkX score and delta score settings). In our discussion of Figures 4A-C showing the XlinkX benchmarking (Results sub-section “Benchmarking Scout on independent XL-MS standard datasets”), we described our rationale behind using the static and dynamic score cut-off in more detail.

21. Line 266 - It is interesting to see that, although the FDR is impressively low in both Scout and xiSEARCH PPI identifications, there is a relatively small amount of overlap. It would be useful to have more discussion on why this may be and pointing out that the softwares can be very complementary as a discussion point - what factors about the spectra, peptides or proteins (for example, abundance or more complex modified peptides) are each of these programs missing? Another reason could be that they are they missing well-formed doublets required for Scout identification, but not for xiSEARCH identification. Some analyses of these Scout-unmatched spectra would be very useful and informative.

We are indebted to the reviewer for asking us to revisit this plot, as it revealed a critical error in our initial analysis. Specifically, we did not take two factors into account:

- 1) Xi reports the cross-linked proteins in a different order than Scout (e. g. Scout: P1-P2, xiSearch: P2-P1). Different from Scout, Xi does not (always) adhere to the numerical order of reporting the first protein as P1, just like Scout does. Therefore, some overlapping PPIs were reported differently and, thus, counted as different PPIs.*
- 2) Ambiguous PPIs identified by Xi (e.g., P1;P2-P3) were listed separately from the unambiguous identification of the same PPI (e.g. P1-P3) and therefore counted as different PPIs.*

After taking these points into consideration, we obtain a substantially higher and more realistic overlap between Scout and Xi:

Figure R 5. Re-analysis of the overlap of PPI-level identifications from Scout and xiSEARCH using the PPI benchmarking dataset by Lenz et al. and 1% separate software-defined FDR cut-off on PPI level. Scout was operated with standard parameters and xiSEARCH identifications were retrieved from the original publication. Percentage numbers indicate empirical FDR determined using the procedure suggested in the original Lenz et al. publication

Action taken: We have replaced Figure 6C by Figure R 5.

22. More general comments:

Is there compliance with export of data in mzIdentML 1.3.0, for PRIDE submissions?

Scout supports mzIdentML 1.2.0 and, with its latest release (<https://github.com/diogobor/Scout/releases/tag/1.5.0>) also mzIdentML 1.3.0 (see <https://github.com/diogobor/Scout?tab=readme-ov-file#data-files>), making it the first search engine for cleavable cross-linkers whose output is compatible with a full PRIDE submission. We have worked closely with the PRIDE team to implement these formats. Since mzIdentML 1.3.0 has not yet been officially released by the HUPO Proteomics Standards Initiative (see <https://www.psidev.info/mzidentml>, last accessed on May 9th, 2024) and is not yet accepted by the PRIDE submission tool, we are only mentioning mzIdentML 1.2.0 in the manuscript. Should mzIdentML 1.3.0 get officially released before acceptance of this manuscript, we will make sure to mention support of this format in the Scout methods section.

23. The authors need to more explicitly describe which type of MS analyses are compatible, and also performed at different steps. It seems here that software was trained on, and analyses stepped-HCD MS/MS of DSSO cross-links. Does it work for other MS-cleavable cross-linkers? Does it work for MS3 analysis? How well does it generalise - e.g. the efficiency of doublet generation is different across different crosslink molecules.

The reviewer's description is correct. While we recognize that different benchmarking datasets with all possible MS-cleavable cross-linkers and all different acquisition strategies would bring benefit, due to the large scale of our XL-MS standard, it was not feasible to exhaustively test different combinations of cross-linkers and MS acquisition strategies in the framework of this study, which is why we chose to make an XL-MS standard with DSSO and use a stepped-HCD MS2 strategy optimized for this type of sample. We made these choices for three reasons:

- I) *DSSO and stepped-HCD were also used in the benchmarking studies by Matzinger et al. (Nat Commun 2022) and Lenz et al. (Nat Commun 2021), making it possible to integrate their published datasets more easily in our study*
- II) *Advantages of stepped-HCD MS2 over MS3 strategies were previously demonstrated on ResPair level using the peptide-based ground-truth dataset from Matzinger et al. (Nat Commun 2022, 10.1038/s41467-022-31701-w - Figure 9)*
- III) *In our experience with biological datasets, stepped-HCD MS2 provides the most comprehensive results in proteome-wide XL-MS experiments with MS-cleavable cross-linkers*

To further support our reasoning for using stepped-HCD MS2, we re-measured batch 2 from our analytical standard using both a stepped-HCD MS2 and a MS2-MS3 acquisition method. Analyzing these data with XlinkX PD (since this software does accept MS3 data) and dynamic score filtering confirmed the superior performance of stepped-HCD MS2 (Figure R 6).

Figure R 6. Cross-link identification numbers when analyzing batch 2 with stepped-HCD MS2 and a MS2-MS3 acquisition method. High-confidence identifications pass the dynamic score filter of XlinkX PD, low-confidence identifications are only found when using the default settings of XlinkX PD without further post-processing. Small and large database refer to the 540-entry and 4000-entry databases, respectively, that were used for the software comparison in the manuscript.

That said, we do acknowledge that these choices have had an impact on the design of Scout, which is currently only compatible with cleavable cross-linkers and does not accept MS3 data. Nonetheless, Scout is compatible with different MS-cleavable cross-linkers as shown in our newly added analysis of biological datasets obtained with Azide-A-DSBSO and TDS (see our response to Reviewer#1's first comment for details on these analyses).

Action taken: We added Supplementary Note 6 to further rationalize our experimental choices for our analytical standard, including our decision for a stepped-HCD MS2 strategy. In the Discussion section, we highlight the limitations of Scout with regard to non-cleavable cross-linkers and MS3 data.

24. Also would be good to have a forward looking discussion of possibility of expanding Scout to other types of crosslinkers.

Please refer to our previous response. Scout is readily applicable to other cleavable cross-linkers, as shown by the newly added analyses of biological datasets. However, Scout's current design is incompatible with non-cleavable cross-linkers.

Action taken: We have added this point to the Discussion.

25. As stated above, this is a very impressive and important study, yet it would be very nice if a completely new “real life” Xlinking dataset would be included in the study in addition to the analysis previously published datasets.

We agree that it is important to demonstrate that Scout adds value when applied to biological XL-MS datasets. Therefore, we have applied Scout to an unpublished deep XL-MS dataset obtained from intact HEK293T cells cross-linked with Azide-A-DSBSO. In addition, we have applied Scout to published XL-MS datasets of human mitochondria and intact HeLa cells generated with different cleavable cross-linkers. The results and corresponding revisions are described in our response to Reviewer#1's first comment.

26. Regarding NN architecture. From what is written in Supp Note 4, the neural network is a canonical Multi-Layer Perceptron. Its main purpose is to capture non-linearities between inputs that are relevant to predict the output. Here, the output is a binary state indicating whether two proteins really interact or not. Overfitting to the training data is the most important aspect to consider in this type of densely connected neural nets. To prevent it, authors use two approaches: 1) Fine tune the model to a subset of 64 proteins in their standard and 2) L2 regularization on neurons weights. I am curious about how these 64 proteins were selected, as this is not indicated in any part of the manuscript. The second point that raises my curiosity is why they do not provide any standard metric about the binary classification performance during training following a cross-validation scheme (e.g. AUROC, or in this case AUPRC if TP/TN ratio is really small).

This question covers several aspects, each of which is addressed below.

Re standard protein selection:

All 256 proteins were randomly allocated to 32 interaction groups, which were then randomly allocated to 4 batches. As such, the 64 proteins (batch 1, 8 interaction groups) represent a random selection from our pool of recombinant proteins.

Re cross-validation:

While cross-validation is a valuable tool in many machine learning contexts for model tuning and validation, applying this approach to MS proteomics data is more challenging because of their sparsity (i.e. many target sequences present in the database are not observed in the sample). This results in a high likelihood of incorrect class labels in the target dataset and makes it difficult to control against overfitting. This problem is exacerbated in XL-MS because true-positive inter-link spectra are even scarcer than true-positive spectrum matches in

traditional proteomics. Taking these issues into account, we opted for an approach that diverges from conventional machine learning practices (i.e. a model, once trained, applies its learned patterns to classify new, unseen data). Instead, we relied on ANNs, whereby their hyperparameter optimization during Scout development was guided by the 64-protein dataset (batch 1), which provided a set of allowed vs. forbidden CSM/ResPair/PPI (according to our controlled mixing scheme). Importantly, ANNs re-train on each new dataset, a process controlled by the pre-determined hyperparameters.

Re binary classification performance:

Even though the ANN training is based on a binary classification system of the inputs (either target or decoy), the ANN generates a continuous numerical output for each cross-linked spectrum match (CSM), rather than a simple binary output. This continuous output is derived from the final layer of the neural network, which uses a logistic function to interpret the raw scores calculated by the network. The output represents Scout's predicted possibility that the CSM is true and allows Scout to rank all CSMs based on their predicted confidence levels. Since all Scout searches are performed against a target-decoy database, the ranked CSMs will either correspond to target matches or decoy matches. The ranking of all results, thus, allows establishing a score threshold that separates true-positive from false-positive identifications with the optimal balance between sensitivity and specificity.

Re overfitting:

To control that no overfitting has occurred, Scout's performance and reliability was validated during the software benchmarking based on independent datasets that were not used for Scout development and ANN hyperparameter optimization (batches 3 and 4 + the Matzinger et al. data + a newly added entrapment experiment shown in Figure R 1A). Reassuringly, the Scout results with these datasets have empirical FDRs that closely match the software-defined Scout FDR setting. This rigorous evaluation strategy confirms Scout's efficacy and adaptability to new datasets.

Reviewer #3:

Remarks to the Author:

This manuscript presents both a new approach to generate a large-scale XL-MS benchmark dataset and a new XL-MS search engine. The scale of the benchmark dataset and performance of Scout even on already existing benchmarks looks good at first glance. The manuscript is well-written but a number of important details are missing that would help properly evaluating it.

We thank the reviewer for their constructive criticism and hope we have adequately addressed their comments.

27. Scout appears better than a number of search engines used in the field although XlinkX and xiSEARCH come close on the PPI level, once FDR is properly controlled.

We agree with this summary, but we wish to point out that comparing Scout-vs-xi identification numbers was only possible when using small/medium-sized protein sequence databases (540 human sequences or 1929 E.coli sequences) and running xi on a computer cluster with 256GB RAM and many computing cores (Figures 4D, 6D and Supplementary Fig. 2A). When using the standard computer setup employed for the other software comparisons to compare the speed of Scout and xi, we could only analyze 4 raw files from our benchmarking dataset using the 540-protein database and because xi wouldn't run to completion with larger datasets and databases. This is in line with xi user experiences reported by Matzinger et al., Nat Commun 2022, 10.1038/s41467-022-31701-w. As such, we believe that Scout will be accessible to more users and applications, because it can analyze complex XL-MS datasets on a standard desktop computer within minutes to hours.

28. Scout is also much faster, however a number of questions and concerns are raised below about the details of performance comparisons.

1) The authors propose that instead of peptide synthesis as described in Matzinger et al, to express whole proteins or fragments in E.coli. It's important to note that both approaches seem to produce similarly useful but unrealistic datasets. Rather than cross-linking purified peptides, here the authors cross-link purified proteins after heating them to induce denaturation/aggregation. Since this has no biological relevance, the main advantage seems to be efficiency/scale of peptide production? It might help the reader to explain the general thinking behind the approach compared to just synthesizing more peptides, considering the more predictable results of peptide synthesis. The nature of "PPIs" produced this way seems about as artificial as mixing subsets of purified peptides. In fact, I would be much more comfortable labeling the data resulting from the benchmark experiment something like "Protein Pairs" throughout the manuscript to avoid confusion with actual interactors in other datasets.

We agree that we failed to discuss the key advantages of our dataset over the Matzinger et al. dataset (beyond its size) in sufficient detail.

Generally speaking, a standard sample and the derived datasets – especially when used for developing and benchmarking of software – should simulate many features of the real-world data that the software is applied to. The key advantages of cross-linking purified protein pairs rather than peptides are summarized in Table R 2 below:

	'real-world' proteome-wide XL-MS	Recombinant protein standard reported here	Synthetic peptide standard by Matzinger et al.
Starting sample	Proteins in solution (concentration range 10^7 in human cells)	Proteins in solution (concentration range 0.4-1.2 mg/ml)	Peptides in solution (always 1 linkable site per peptide) at fixed concentration
Sample measured by MS	Peptides derived from proteins by protease digestion	Peptides derived from proteins by protease digestion	Synthetic peptides
Sample composition	Linear peptides >> Cross-linked peptides from intra-linked proteins >> Cross-linked peptides from inter-linked proteins	Linear peptides >> Cross-linked peptides from intra-linked proteins >> Cross-linked peptides from inter-linked proteins	Cross-linked peptides > Linear peptides, No intra-links, no inter-linked proteins
Data structure (see also our response to comments 14, 16, and 17)	Most MS2 spectra matched to linear peptides, CSMs relatively rare, PPI coverage incomplete	Most MS2 spectra matched to linear peptides, CSMs relatively rare, PPI coverage incomplete	CSMs dominant
Cross-link identification levels captured	CSMs, Unique residue pairs, PPIs	CSMs, Unique residue pairs, PPIs	CSMs, Unique residue pairs

Table R 2. Properties of biological and standard datasets in XL-MS

Importantly, our protein-based dataset allows us to distinguish different types of cross-links (intra-links and inter-links) and assess all levels of cross-link identifications (CSM, ResPair, PPI). This is critical information for developing machine-learning based scoring functions at PPI level to enable robust FDR control. This is impossible with a peptide-based dataset as it only contains CSM and ResPair level information, and does not resemble protein-derived intra-links and inter-links.

We intentionally did not consider biologically occurring PPIs, because this would have limited our abilities to control protein contact formation. Because the dataset generated here are intended for software development and benchmarking on the LC-MS level, the physicochemical and structural basis for cross-link formation is not relevant. Our goal was to generate well-controlled datasets that are sufficiently complex and contain a sufficient number of inter-links to resemble a proteome-wide XL-MS dataset and heat aggregation was the most reliable approach to achieve this (please also refer to our responses to Reviewer#2's comment 14).

Comparing our standard dataset to real-world data, the main limitation is that its pairwise mixing scheme does not fully resemble the complexity and connectivity of a naturally occurring protein interactome. We are aware of this problem and had mentioned it in the Discussion section of the original submission.

We understand the reviewer's concern over the term "PPI". We used this terminology to distinguish the FDR determined on the level of protein pairs from the FDR determined on the level of unique residue pairs and CSMs. The term PPI-FDR has previously been established for the protein pair level (Lenz et al., Nat Commun 2021, 10.1038/s41467-021-23666-z) and we are hesitant to introduce a different term to describe the same concept. That said, we agree that the differences to biological PPIs needs to be clarified in the main text.

Action taken: We included the advantages and limitations of our recombinant standards (summarized in Table R 2) in the Discussion section. We highlighted in the first sub-section of the Results that "PPI" in our standard refers to protein pairs and not to native protein interactions.

29. 2) While the scope of the experiment described in Figure 1 appears impressive, digging into the details, it seems only a fraction of the inter-protein cross-links can be detected. Of the 896 possible protein pairs, 193 are detected in searches of batches 3 and 4, using the 540 fasta, which is presumably the most sensitive. I don't see any information to help figure out what fraction of the overall dataset this represents, but since $\frac{1}{4}$ of the dataset was used for training, and there are 4 batches mentioned, I would guess this represents $\frac{1}{2}$ the dataset? If that's the case, the authors are detecting ~40% of possible cross-linked protein pairs? This would be an important metric to report.

The reviewer's calculation is by and large correct. As mentioned in Table R 2 above and discussed in more detail in our response to Reviewer#2's comment 14, we thus believe that the incomplete PPI coverage provides additional evidence that our standard datasets resemble important features of 'real-world' proteome-wide XL-MS data. Of note, the number of detected PPIs strongly depends on the search parameters and software. For example, a less stringent FDR cut-off will result in the detection of more true PPIs but these will be interspersed with a greater number of false PPIs that are forbidden according to our mixing scheme. Similarly, consideration of additional cross-linking sites (e.g. S, T, Y in addition to K) will increase the number of detected PPIs.

Action taken: The incomplete PPI coverage is now discussed in the Results section when we describe the results of Figure 3.

30. One possible explanation for such relatively low rate could be that proteins are purified "at concentrations varying from 0.1–1.5 mg/mL" That's a pretty wide range, and it would make sense that some of the protein pairs mixed at < 0.5 mg/mL would be much less likely to produce inter-links, even after heating. However, I don't see any such info for specific proteins, so it's hard to confirm/deny such hypotheses. Another possibility is that some fraction of proteins aggregate and precipitate out of solution due to heating before they have a chance to get cross-linked.

We thank the reviewer for this comment. Although we were unable to reconstruct the naturally occurring dynamic range of a complex biological sample, we intentionally varied the protein concentrations in our standard to generate a more realistic dataset. This is another important difference to the Matzinger et al. approach (see also Table R 2). As shown in Figure R 7, the number of detected inter-links neither correlates with the protein concentration, nor with the number of Lys residues within the protein. We believe that the aspects discussed in response

to the previous comment 29 (relative sparsity of inter-links and low abundance of cross-links) are more likely the reasons that some protein pairs are not detected.

Figure R 7. Number of detected inter-linked residues plotted against number Lys residues in the respective protein and protein concentration. Batches 1-4 represent the 4 datasets generated from our analytical XL-MS standard, whereby each dataset comprises 8 interaction groups (=64 proteins).

31. 3) While it is stated that 256 proteins “were provided by Absea Biotechnology”, it seems a significant fraction of them might not have expressed, or at least they are not listed in the Supplementary Data 1 pdf. This would not be surprising in such a large-scale operation, but it would be helpful to include such details in the Methods somewhere, rather than keep the readers guessing. This would, in turn, help adjust expectations of the max # cross-linked protein pairs we should expect to see.

We apologize for this mistake. We accidentally uploaded an earlier version of this file, which did not include all protein data. We can assure the reviewer that all protein constructs have been successfully expressed; in fact, they are all commercially available now (see <https://www.absea.bio/product-proteins-soluble.html>).

Action taken: We updated Supplementary Data 1 to include all protein data.

32. 4) The details of how exactly the proteins were mixed needs to be included in supplementary materials with the paper, not just in PRIDE.

Action taken: We have included information on the composition of the interaction groups and dataset batches in Supplementary Data 1.

33. 5) According to Suppl. Table 2, the authors for some reason have chosen to use KSTY reactive sites for Scout and XiSEARCH but not for other search engines. This seems like a very strange choice – this is clearly an available option for most if not all of them (MS Annika, XlinkX and MeroX were all compared with K vs KSTY in the Matzinger et al paper suppl.). This parameter may make a difference depending on the specifics of the search engine and/or sample prep conditions. If KSTY searches from Matzinger are not available or obtainable, it would make sense to just use K-K for Scout as well for a fair comparison in relevant figure/tables. This absolutely must be addressed.

We apologize for this misunderstanding. Please refer to our response to Reviewer#1's comment 7. In brief, KSTY specificity was only used for comparing Scout to xi, because this

setting was recommended by a xi co-developer. For all other comparisons, all tools including Scout were run with K-only specificity.

Action taken: We have clarified in Supplementary Table 2 that KSTY specificity was only selected for the Scout-vs-xiSEARCH/xiFDR comparison and that Scout was limited to K specificity for all other comparisons. We also highlighted this in our description of the Scout-vs-xi comparison in the main text.

34. 6) What is the ANN implementation used? GNB? Did the authors write their own or rely on existing libraries? The number of input parameters used is pretty small so simpler discriminant approaches might have worked just as well, while also providing insight into parameter weights, rather than the ANN black box. This would be interesting to help figure out if the improved performance of Scout is primarily due to a large training set or the input scores, which don't appear to be particularly novel.

Artificial Neural Networks (ANN) and Gaussian Naïve Bayes (GNB) are two different machine learning methods, whereby GNB is not a type of neural network. Scout relies on ANNs and, for ANN implementation, we selected the multilayer perceptron (MLP) from Python's scikit-learn library, as it is a widely adopted solution and thoroughly tested in the machine learning community (see Supplementary Note 4). We did preliminary tests with several discriminant approaches, ranging from the simple Naïve Bayes to the elaborate support vector machines, but we found that MLP was consistently the most stable and efficient solution.

*Considering that existing XL-MS search engines already rely on very sophisticated scoring schemes, we did not feel that their shortcomings can be addressed by introducing additional scores. Therefore, the novelty of Scout lies not mainly in its scores but in the way they are handled and employed by the software, allowing Scout to efficiently deliver FDR-controlled proteome-wide XL-MS results within an hour, utilizing minimal resources (16GB RAM). The advantages of Scout arise from thoroughly optimizing details of the methodology of fragment match collection, ion selection in theoretical spectra, spectrum pre-processing (e.g. employing the specialized deconvolution algorithm Y.A.D.A. 3 developed by us – Clasen et al. *Bioinformatics* 2022, 10.1093/bioinformatics/btac638), and ion intensity handling. These seemingly minor changes are crucial because minimal modifications of these aspects can significantly impact identification accuracy and the filtration of decoy hits.*

Finally, it is important to clarify that the efficacy of Scout does not stem from an expansive training set. Rather, the 64-protein-dataset served as a foundational dataset to optimize the ANN hyper-parameters during Scout development (see also our response to Reviewer#2's comment 26). There was no transfer of learning between this dataset and the datasets used for software benchmarking, ensuring the integrity and independent validation of our approach.

35. 7) Authors mention that a portion of the dataset was used for training, and another, independent portion for testing, but it's not clear specifically which portions were used for what purpose and how big they are in terms of number of proteins/pairs. The search results in PRIDE appear to refer to 3rd and 4th batches, but it's not clear what those refer to. Are the search results for 1st and 2nd batches missing in PRIDE?

We apologize for not explaining use of our data more clearly. The analytical standard (256 proteins allocated into 32 interaction groups) was split into 4 dataset batches, with each batch comprising 8 interaction groups. The dataset used to guide the development of Scout was is

named “first batch” in the PRIDE submission. The benchmarking dataset used for software comparison is based on batches 3 and 4, thus comprising 128 proteins in 16 interaction groups (third and fourth batch in the PRIDE submission). The remaining dataset (second batch in the PRIDE submission) was used for optimizing our LC-MS method in preparation of this study. We originally intended to include batch 2 also in the benchmarking dataset, but we encountered technical issues during the final MS experiment, which meant that batch 2 - while confirming the results obtained with batches 3 and 4 - did not add substantial value to the software benchmarking and hence was not used for this analysis. We apologize for not explicitly stating this in the initial submission. We ultimately decided to keep batch 2 excluded from the software comparison, mainly for the following reasons:

- I) While we could have included batch 2 in the comparison of Scout vs. MS Annika, XlinkX PD, MeroX and MaxLynX, we would not have been able to include batch 2 in the Scout-vs-xi comparison. The reason is that running xi on such a complex dataset is only feasible on a computer cluster – xi would not run to completion on our in-house desktop computers (as explained in response to Reviewer#1’s comment 7 and this reviewer’s comment 27). We could get access to an external computer cluster through a collaborator for the initial Scout-vs-xi comparison (based on batches 3 and 4), but we were not able to get access to a computer cluster in the time frame of this revision. We felt it would not be ideal to include batch 2 in some but not all software comparisons.
- II) We initially used batch 2 for LC-MS method optimization and, during this revision, we used this batch again for LC-MS method comparisons (stepped-HCD MS2 vs. MS3 based acquisition strategies) in response to Reviewer#2’s comment 23. In these experiments, either Scout or XlinkX PD were used as read-out tools. Therefore, we feel that excluding batch 2 from the software comparison is the “cleaner” solution for our study, because it ensures full independence of the benchmarking data and makes the software comparison fairer.
- III) We have shown that already a single batch has the MS2-level complexity of a proteome-wide XL-MS experiment (see our response to Reviewer#2’s comment 16). As such, batches 3 and 4 certainly offer sufficient depth and complexity to assess the search engines’ performance for proteome-wide XL-MS. Therefore, including batch 2 in the software comparison is not essential.

Importantly, the development of Scout and the XL-MS search engine benchmarking are just two application examples for the standard datasets that existed at the time of writing. We see this recombinant standard and the resulting datasets as a living and expanding project, conceptually similar to ProteomeTools. With this in mind, we designed our standard to be easily expandable by adding new interaction groups. Over the last months, we already generated a 5th standard dataset with 8 additional interaction groups. Future users of these datasets can freely re-use and re-combine them for their own software development or benchmarking studies. To support this, we have made all standard datasets available as a separate, full, DOI-minted PRIDE submission.

Action taken: We have made two PRIDE submissions: (1) a partial submission containing all raw data used in this manuscript as well as all search results underlying the main and supplementary display items (partial because the other tested XL-MS search engines don't support the output formats that would be required for a full submission); (2) a full submission containing all raw data currently available from our recombinant standard (5 batches in total). We added a paragraph to the first Results sub-section ("Constructing an XL-MS standard mimicking complex biological samples") to clearly describe which batch was used for which purpose. The composition of the individual batches and interaction groups is now indicated in Supplementary Data 1.

36. 8) Line 147: "For intra-links, ... the most relevant information comes from the ResPair level, for which Scout shows the best overall performance (Supplementary Figure 1)." I don't disagree with this statement, but I think it would be interesting to try to understand why Scout was outperformed (e.g. by MSAnnika) by up to ~50% on CSM level, and by ~20% on the protein level, but not on the ResPair level! Could this have something to do with how FDR is controlled, or the ratio of inter/intra links in the training dataset? It is likely Scout would be used by the community for small-scale datasets and single proteins as well, so it's important to understand how it performs in different circumstances and why. Again, it's not clear which portions of the dataset are used here. The CSM numbers don't appear to quite line up with the identifications_thirdbatch_fourthbatch_allsearchengines file in PRIDE.

We agree that Scout would and should also be used for small-scale datasets, which is the main reason why we also benchmarked it on a ground-truth dataset from Matzinger et al. (Nat Commun 2022), which only includes 100 synthetic peptides. As shown in Supplementary Table 3, Scout performed very well on this small-scale dataset.

The software comparison for intra-links was done based on the datasets from batches 3 and 4 (same as the comparison for inter-links).

We agree that it would be interesting to understand the reasons for the performance differences between Scout and MS Annika. Unfortunately, the publication of MS Annika only provided pseudo-code (Pirklbauer et al. JPR 2021, 10.1021/acs.jproteome.0c01000) and the source code is not available, so we can only speculate which features MS Annika might have. We agree with the reviewer that differences in the FDR control strategies are very likely to contribute to the observed differences. Unlike Scout, MS Annika employs a conventional FDR estimation strategy. It first applies a target-decoy FDR cut-off to all identified CSMs, followed by aggregation of the remaining target and decoy CSMs and their scores to ResPair cross-links, on which another target-decoy FDR cut-off is applied. Protein/PPI-level scoring and FDR control are not part of MS Annika, which is why we had to revert to post-hoc aggregation to obtain results on this level. MS Annika's scores rely solely on contributions from spectral features, which remain impactful across all identification levels. By contrast, in Scout only the calculation of the CSM Classification Score is predominantly influenced by spectral features. Importantly, Scout uses ANN-based score calculation on every level (CSM, ResPair, PPI), influenced by different features unique to each level. This results in different stringency and aggregation behaviour from lower to higher levels compared to conventional software.

The main focus during Scout development (which was guided by the batch 1 dataset, as described in the Supplementary Notes) was to achieve rigorous FDR control for PPIs. In all our analyses, MS Annika identified more PPIs than Scout but showed a greater propensity of

false-positive PPI identifications, probably also because it does not feature a dedicated PPI-level FDR control. The chance of false discoveries is much lower for intra-links than for inter-links. As such, the good performance of MS Annika on intra-linked protein level might be explainable by its more permissive FDR filtering, which in the case of intra-links won't lead to a substantial increase in false identifications.

Moreover, we do not know whether or not MS Annika employs boosting techniques that leverage identified proteins to infer less confident identifications related to those proteins or protein pairs. Scout's default setting does not incorporate boosting. This decision is based on our observation that such strategies, unless meticulously managed, can inadvertently elevate the rate of false identifications for a given protein, potentially bypassing the FDR strategy employed in our study.

Since this discussion is highly speculative, we would prefer to omit it from the manuscript. In our view, the most important message from our benchmarking studies for future users of Scout is that the identifications made by this software are highly likely to be true and the set FDR cut-off will be very close to the actual fraction of false identifications. Our analyses have shown that this important feature holds true for small and large protein sequence databases, simple and complex cross-linked samples, inter-links and intra-links.

37. 9) In Figure 4C, Scout appears to outperform XlinkX by a wide margin on CSM level, and even wider margin on ResPairs level, but slightly under-performing (540 fasta) on the PPI level. This is a bit surprising and some discussion of possible causes is warranted. E.g., Scout vs xiSearch profile makes a lot more sense with gradually decreasing outperformance from CSM to PPI level.

It is correct that XlinkX scoring functions are very well tuned to sensitively detect PPIs in small databases, and this indeed allows XlinkX reporting a slightly higher number of PPI using the small 540 protein sequence database. XlinkX's main shortcoming, however, is the lack of a robust FDR control. Currently, the only way to address this issue within XlinkX is to set heuristic score cut-offs, but these are difficult to optimise in real-world data where no ground-truth is available. In other words, in a 'real-world scenario' a result as shown in Figure 4C would not be possible, because there would be no way to filter the results to an empirical FDR of 1%. Instead, the best possible outcome would be the one shown in Figure 4B, where – despite applying a setting the FDR to 1% in XlinkX and applying a dynamic score cut-off (which is rarely ever done) – over 4% of the identified PPIs will be false-positives.

This phenomenon further illustrates the importance of the benchmarking dataset generated in our study, which allowed us to identify the challenges of FDR control in XlinkX and precisely evaluate the empirical FDR when applying different XlinkX score cut-offs.

Action taken: We have included more detailed comments on the advantages and limitations of XlinkX and the other search engines in the Discussion section.

38. 10) I downloaded the latest release of Scout. The interface looks nice and is fairly intuitive, but when I tried running it with default params on one of the RAW files in PRIDE. The result was the following error:

```
ERROR - PerformSearch method: It's not possible to run the search.
at System.Linq.ThrowHelper.ThrowArgumentOutOfRangeException(ExceptionArgument
argument)
at System.Linq.Enumerable.Chunk[TSource](IEnumerable`1 source, Int32 size)
at ScoutCore.PSMEngines.Tripper.Tripper.RunSpectraSearchParallel(List`1 allQueries,
List`1 allPeptides)
at ScoutCore.PSMEngines.Tripper.Tripper.PerformSearch(String fastaFile, String
rawFile)
Specified argument was out of the range of valid values. (Parameter 'size')
```

We are very sorry that Scout did not work in the reviewer's hands and thank for sharing the error message. The error message suggests that the reviewer was running Scout on a virtual machine. Another reason might have been to set up only one single core in their virtual machine. As mentioned on Scout's GitHub page (<https://github.com/diogobor/Scout>), we recommend running Scout on a Windows platform (see software section) and using 4 computing cores (see hardware section). We have now updated Scout (<https://github.com/diogobor/Scout/releases/tag/1.5.0>) to run more stably on virtual machines provided that the hardware requirements are met; however, our primary recommendation remains to run Scout on Windows platforms.

REVIEWER COMMENTS

General note: We have numbered all reviewer comments consecutively to make this document easier to navigate

Reviewer #1:

Remarks to the Author:

As already mentioned in the first review round, Scout seems an impressively fast and well working software, even further improved now, which will be of great value to the community. The authors carefully and comprehensively answered all reviewer questions and significantly improved the quality of their manuscript by adding a novel DSBSO based proteome-wide dataset. That said I would like to congratulate the authors to their excellent work and recommend publication of a revised version.

We are very grateful for the reviewers constructive and supportive comments that substantially contributed to increasing the quality of our manuscript.

Here are some minor comments:

1. I appreciate the addition of Table R2 comparing used benchmarks to real datasets. Especially given the differences in dynamic range and ratio of linear peptides to crosslinked peptides. In that table the authors mention the Matzinger et al dataset based on synthetic peptides. To the best of the reviewer's knowledge Matzinger et al digested the synthetic peptides using trypsin and performed a reduction step to remove protecting parts and obtain tryptic peptides as would be present from a standard digest. Therefore, I would suggest something like adding "Synthetic peptides, final peptide obtained by protease digestion" (or similar) to the table to clarify the exact design (which is not as different as it seems in the current table).

We thank the reviewer for spotting this ambiguity and agree with the correction.

Action taken: Since this table is not included in the manuscript, we are pasting the corrected version below (amendments are underlined) and make it available to the readers by participating in Nature Methods' Transparent Peer Review process.

2. Regarding the question of PPI from the Matzinger et al. dataset I have the following thought: It's true that there are no real PPIs but from the perspective of the data analysis software there are PPIs since the used synthetic peptides correlate to 38 different proteins therefore yielding inter-protein crosslinks in the search. I would recommend adopting the text in Table R2 accordingly.

We apologize that the phrasing was not sufficiently clear. The point we were trying to make is that the Matzinger et al. standard is not designed for assessing identifications on the PPI level because there are too few interacting proteins and they are only represented by selected peptides. At the same time, we fully agree that some of the inter-linked synthetic peptides represent different proteins.

Action taken: To make these two aspects clearer, we changed the description of the Matzinger standard in the “sample composition” category and revised the name of the final category to “testable cross-link identification levels”. The amendments are underlined in the table below.

	‘real-world’ proteome-wide XL-MS	Recombinant protein standard reported here	Synthetic peptide standard by Matzinger et al.
Starting sample	Proteins in solution (concentration range 10⁷ in human cells)	Proteins in solution (concentration range 0.4-1.2 mg/ml)	Peptides in solution (always 1 linkable site per peptide) at fixed concentration
Sample measured by MS	Peptides derived from proteins by protease digestion	Peptides derived from proteins by protease digestion	Synthetic peptides, final peptide obtained by protease digestion
Sample composition	Linear peptides >> Cross-linked peptides from intra-linked proteins >> Cross-linked peptides from inter-linked proteins	Linear peptides >> Cross-linked peptides from intra-linked proteins >> Cross-linked peptides from inter-linked proteins	Cross-linked peptides (some representing inter-linked proteins) > Linear peptides,
Data structure	Most MS2 spectra matched to linear peptides, CSMs relatively rare, PPI coverage incomplete	Most MS2 spectra matched to linear peptides, CSMs relatively rare, PPI coverage incomplete	CSMs dominant
Testable cross-link identification levels	CSMs, Unique residue pairs, PPIs	CSMs, Unique residue pairs, PPIs	CSMs, Unique residue pairs

3. In line 1 I would recommend a similar rephrasing in the manuscript text, i.e. line 301 in the revised manuscript, could be rephrased to something like “Matzinger et al contains only limited PPI based information” (or similar).

Action taken: We agree and have changed the manuscript text accordingly.

4. 2. In Figure 2 I would recommend to add as possible input “.d” MS2 files as well, since this is supported by Scout.

Action taken: We have changed the figure as suggested.

5. 3. The newly added HEK-DSBSO dataset shows impressive coverage with ~2000 PPIs found. I am assuming that crosslinks were predominantly found from abundant protein complexes within the HEK cells. However, since the coverage is that excellent, I was wondering if the authors also found something less abundant in there? (Compare to Nouchikian et. al., <https://pubs.acs.org/doi/10.1021/acs.analchem.3c04682> who reported crosslinks can only be found at the top 20% most abundant proteins.) Where there crosslinks from all cellular compartments including the nucleus present?

Nouchikian et al. suggest, as a general guideline, that if a protein of interest is not among the top 20% most abundant proteins, the likelihood of detecting cross-link substantially decreases. We aimed to maximize the cross-link and interactome coverage by performing Azide-A-DSBSO enrichment, SEC, and high-pH fractionation.

To answer the reviewer's question, we compared our ResPair-level cross-links to HEK cell protein abundances we determined in-house using iBAQ. The results shown below suggest that our enrichment strategy expanded cross-link coverage beyond the 50% most abundant proteins:

As a result, we were able to identify protein-protein interactions (proteins containing inter-protein cross-links) in all major cell compartments, including the nucleus, mitochondria, endoplasmic reticulum, and cytoplasm (subcellular localization annotations extracted from uniprot). Proteins with multiple compartment annotations are counted in each compartment.

6. 4. Furthermore, I was wondering how many cells/how many dishes were used in total to obtain the aforementioned results from HEK+DSBSO? I could only find the protein concentration of 10mg/mL but would be curious about the cell# needed for the described workflow involving the orthogonal bead based + SEC based enrichment. This would be of interest also for the community to estimate how much material is lost during enrichment.

We apologize for the lack of details in the Methods section. In this experiment, we used c.a. 20 mg of input material, which corresponds to approximately ten 15 cm dishes at > 80% confluency (ca. 8×10^7 cells). Following DBCO-enrichment, we obtained around 600 μ g of DSBSO-modified peptides. These peptides were applied to SEC chromatography. By pooling only the early fractions containing species of higher molecular weights, we obtained approximately 150 μ g peptides. These peptides were then subjected to high-pH fractionation.

Action taken: We have added information on the amount of input material to the Methods section.

Reviewer #2:

Remarks to the Author:

I am happy with the revision and recommend publication

Mikhail Savitski

Thank you very much for your critical feedback and support!

Reviewer #3:

Remarks to the Author:

The revised manuscript is much improved and many of the crucial missing details have been filled in. I believe it is essentially ready for publication with a few fairly minor issues:

We thank this reviewer for their helpful comments and greatly appreciate their endorsement of this article.

7. 1) Because of the benchmarking nature of this work, the data is likely to be used by the community quite a bit, so it would be useful to understand a bit more about the raw files in PRIDE. While there's a Methods sub-section describing SCX, it's not actually clear where this was used. Were the synthetic benchmark samples fractionated with SCX? The number of raw files varies ~25-40 between different batches – is that because of different fraction pooling? Or were these not fractionated at all and each raw file is a result of pooling some number of mixed protein groups? Maybe an excel file with raw file metadata would be useful for people so that they know what they are searching.

The reviewer is correct that the benchmarking peptides were separated by SCX and the varying number of RAW files is due to the different combinations of SCX fractions.

Action taken: We have included a table with more detailed information on each individual RAW file as Supplementary Data 2.

8. 2) Supplementary Notes 1-4 now provide quite a bit of detail on the implementation of Scout which might be useful to some readers. However, as the authors themselves explain in the rebuttal (36): “The advantages of Scout arise from thoroughly optimizing details of the methodology of fragment match collection, ion selection in theoretical spectra, spectrum pre-processing ..., and ion intensity handling. These seemingly minor changes are crucial because minimal modifications of these aspects can significantly impact identification accuracy and the filtration of decoy hits.” On the software side, the objective of an academic manuscript ideally shouldn’t be just to act as an advertisement for software downloadable elsewhere, but to also try to convey what was learned during its development. Since the software is not open source, it’s important to make sure that such details the authors consider crucial above are included in these supplementary notes.

All details of Scout are now available, since we have released the source code at <https://github.com/diogobor/Scout/releases/tag/1.4.14> (also accessible via <https://github.com/theliulab/Scout>).

9. 3) Line 497 - “high-pH separated enriched phospho peptides”
Since there's no phospho-peptide analysis in this manuscript, I assume this to be copied/pasted from unrelated manuscript?

We thank the reviewer for pointing out this error. This sentence was indeed a remnant from an experiment not included in this manuscript.

Action taken: We have removed this sentence from the manuscript.